# Simple biophysics underpins collective conformations of the intrinsically disordered proteins of the Nuclear Pore Complex

Andrei Vovk[1†], Chad Gu[1†], Michael G Opferman[2‡], Larisa E Kapinos[3], Roderick YH Lim[3], Rob D Coalson[4], David Jasnow[2], Anton Zilman[1,5*]

[1]Department of Physics, University of Toronto, Toronto, Canada; [2]Department of Physics and Astronomy, University of Pittsburgh, Pittsburgh, United States; [3]Biozentrum and the Swiss Nanoscience Institute, University of Basel, Basel, Switzerland; [4]Department of Chemistry, University of Pittsburgh, Pittsburgh, United States; [5]Institute for Biomaterials and Biomedical Engineering, University of Toronto, Toronto, Canada

**Abstract** Nuclear Pore Complexes (NPCs) are key cellular transporter that control nucleocytoplasmic transport in eukaryotic cells, but its transport mechanism is still not understood. The centerpiece of NPC transport is the assembly of intrinsically disordered polypeptides, known as FG nucleoporins, lining its passageway. Their conformations and collective dynamics during transport are difficult to assess in vivo. In vitro investigations provide partially conflicting results, lending support to different models of transport, which invoke various conformational transitions of the FG nucleoporins induced by the cargo-carrying transport proteins. We show that the spatial organization of FG nucleoporin assemblies with the transport proteins can be understood within a first principles biophysical model with a minimal number of key physical variables, such as the average protein interaction strengths and spatial densities. These results address some of the outstanding controversies and suggest how molecularly divergent NPCs in different species can perform essentially the same function.

*For correspondence: zilmana@physics.utoronto.ca

[†]These authors contributed equally to this work

Present address: [‡]Physics Department, Temple University, Philadelphia, United States

Competing interests: The authors declare that no competing interests exist.

## Introduction

Nuclear Pore Complexes (NPCs) are biological 'nanomachines' that conduct all the transport between the nucleus and the cytoplasm in eukaryotic cells. NPCs participate in a vast number of regulatory processes in the cell, as well as pathological conditions such as viral disease and cancer (*Dickmanns et al., 2015*). Transport through the NPC is fast, highly selective and robust with respect to molecular noise and structural perturbations. Transport of relatively small cargoes up to several nanometers in size, or approximately 30–40 kD, occurs by pure diffusion, without specific interactions with the NPC constituents. By contrast, transport of larger macromolecules, such as import of transcription factors and export of mRNA particles is tightly controlled by the NPC. For efficient transport, macromolecules larger than several nanometers in size must be shuttled through the NPC by soluble nuclear transport proteins from a highly conserved family, known as Karyopherins (Kaps) in yeast or Importins/Transportins in vertebrates. Remarkably, despite its high selectivity and efficiency, the NPC does not consume metabolic energy during transport and does not possess an obvious 'gate' opening or closing during transport (*Terry and Wente, 2009*; *Wente and Rout, 2010*; *Stewart, 2007*; *Feldherr and Akin, 1997*; *Mohr et al., 2009*).

**eLife digest** Animal, plant and fungal cells contain a structure called the nucleus, inside which the genetic material of the cell is stored. For the cell to work properly, certain proteins and other molecules need to be able to enter and exit the nucleus. This transport is carried out by pore-like molecular "devices" known as Nuclear Pore Complexes, whose architecture and mode of operation are unique among cellular transporters.

Nuclear Pore Complexes are charged with a daunting task of deciding which of the hundreds of molecules it conducts per second should go through and which should not. Small molecules can pass freely through Nuclear Pore Complexes. However, larger molecules can only pass through the pore efficiently if they are bound to specialized transport proteins that interact with the proteins – called FG nucleoporins – that line the pore. A unique feature of the FG nucleoporins is that, unlike typical proteins, they do not have a defined three-dimensional structure. Instead, they form a soft and pliable lining inside the Nuclear Pore Complex passageway.

Exactly how interacting with transport proteins affects the structure and spatial arrangements of the FG nucleoporins in a way that allows them to control transport is not well understood. This is in part because existing experimental techniques are unable to study the structures of the FG nucleoporins in enough detail to track how they change during transport. The complexity and the diversity of the FG nucleoporins also make them difficult to model in detail.

Vovk, Gu et al. have developed a theoretical model that is based on just three basic physical properties of the FG nucleoporins – their flexibility, their ability to interact with each other, and their binding with the transport proteins. Future work can refine the model by incorporating further molecular details about the interactions between FG nucleoporins and transport proteins.

The predictions made by this simple model agree well with experimental results in a wide range of situations – from single molecules to complex spatial assemblies. They also explain why some of the experimental results appear to contradict each other and suggest how several outstanding controversies in the field can be reconciled. Because the model invokes only fundamental physical principles of FG nucleoporin assemblies, it shows that some of their general properties do not depend on the exact conditions. In particular, this might shed light on why Nuclear Pore Complexes in different organisms perform essentially the same function, although the details of their molecular structure may differ. This also suggests how the FG nucleoporins can be manipulated to build artificial devices based on the same principles.

The spatial organization of the NPC and its transport mechanism are unique. The passageway through the nuclear envelope of about 35–50 nm in diameter and $50 - 80$ nm in length is formed by a structural scaffold that comprises multiple proteins of a combined size of $\sim 150$ MDaltons. This passageway is lined by a set of $\sim 200$ intrinsically disordered polypeptide chains, collectively known as 'FG nups' due to the large numbers of Phenylalanine-Glycine (FG) repeats in their sequence (*Terry and Wente, 2009*; *Wente and Rout, 2010*). Although the actual sequences of the FG nups can vary widely among different species, the overall structure, organization and the transport mechanism of the NPC are conserved (*Terry and Wente, 2009*; *Hülsmann et al., 2012*; *Schmidt and Görlich, 2015*). As the key component of the NPC transport mechanism, the FG nups set up the permeability barrier that prevents free passage of large macromolecules and serve as a template for the transient binding of the cargo-carrying transport proteins. NPCs are also remarkably resilient with respect to structural perturbations: many of the FG nups can be genetically deleted without impairing cell viability and without major effect on transport (*Strawn et al., 2004*; *Popken et al., 2015*; *Feldherr et al., 2002*; *Hülsmann et al., 2012*). Cargo-carrying transport proteins bind to the FG nups through multiple, yet relatively weak contacts. This binding is crucial for selective transport: interfering with it decreases the transport efficiency, or abolishes the transport altogether. Conversely, particles or molecules that normally cannot penetrate the NPC can be transported after chemical modifications that enables the to interact directly with the FG nups (*Wente and Rout, 2010*; *Bayliss et al., 1999*; *2000*; *Naim et al., 2009*; *Kim et al., 2013*; *Kumeta et al., 2012*). NPC geometry and architecture are schematically illustrated in *Figure 1*.

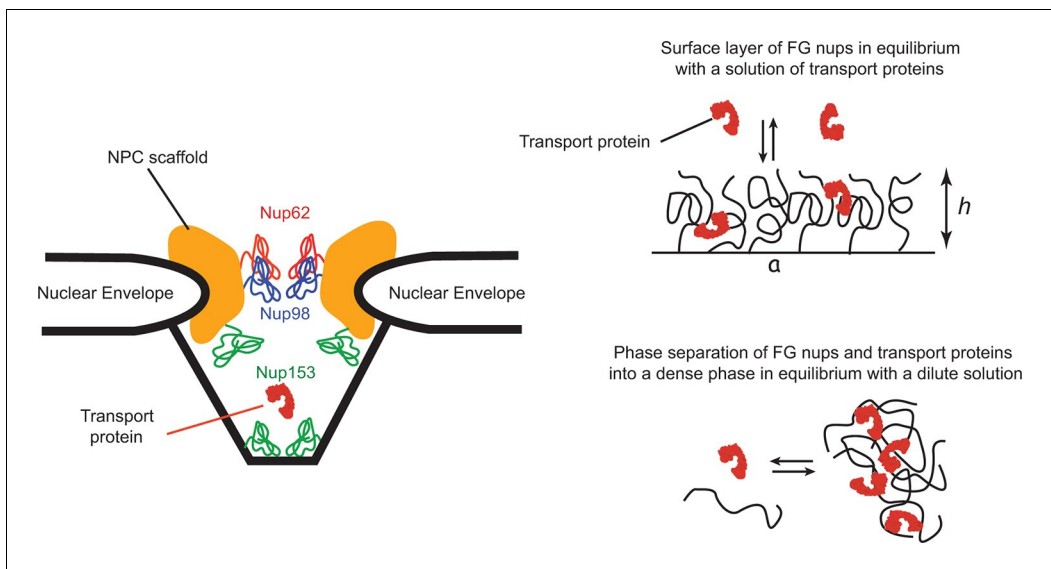

**Figure 1.** Schematic illustration of spatial arrangements of the FG nups in the NPC and in vitro models. *Left*: schematic rendering of the NPC geometry (not to exact scale). The vertebrate FG nucleoporins discussed in this paper (Nup62, Nup98 and Nup153) and their approximate locations within the NPC are highlighted in color (*Chatel et al., 2012*; *Krull et al., 2004*; *Chug et al., 2015*). Other FG nups are not shown. Yeast NPC has an overall similar architecture but smaller dimensions. Vertebrate FG nucleoporins discussed in the paper have yeast analogues: Nsp1 is analogous to Nup62, while Nup100 and Nup116 are analogous to Nup98 in their sequence and the biophysical and functional properties. *Right upper panel*: schematic depiction of one typical in vitro experimental setup of a grafted FG nup layer in equilibrium with a solution of transport proteins. *Right lower panel:* some FG nups, such as Nup98, phase separate at high concentration and form a dense phase in equilibrium with a dilute solution.

Full understanding of the NPC transport still remains elusive, and various hypotheses remain unsettled. The consensus is that the binding of the transport proteins to the FG nups enables them to overcome the permeability barrier. The strength of this binding controls the transport selectivity and efficiency. Hence, transport proteins can be informally viewed as 'glorified enzymes' that lower the free energy barrier for the translocation through the NPC. Basic models that describe the transport as facilitated diffusion through the FG nup medium, modulated by the interactions with the FG nups, provide a good explanation of the selectivity of the NPC even in the presence of large amounts of molecular noise (*Wente and Rout, 2010*; *Zilman et al., 2007*; *2010*; *Frey and Görlich, 2007*; *Fernandez-Martinez and Rout, 2012*). The overall veracity of these general principles has been demonstrated by creation of artificial nanochannels and nanomaterials that mimic NPC function and recapitulate many of its transport properties (*Zilman et al., 2007*; *2010*; *Frey and Görlich, 2007*; *Schmidt and Görlich, 2015*; *Zilman, 2009*; *Jovanovic-Talisman et al., 2009*; *Kowalczyk et al., 2011*; *Caspi et al., 2008*; *Jovanovic-Talisman et al., 2014*).

Various models of the mechanistic involvement of the FG nups in transport have been proposed. In the 'virtual gate' model, the permeability barrier arises due to the entropic repulsion from the fluctuating FG nup chains (*Zilman et al., 2007*; *Lim et al., 2007*; *Rout et al., 2003*). In a related idea, an entropically stabilized FG nup 'brush' can be collapsed by the transport proteins thus opening the transport passageway (*Lim et al., 2006*; *2007*; *2008*). In another scenario, the permeability barrier arises from a gel-like network, stabilized by the hydrophobic interactions between the FG repeats. Transport proteins disentangle this gel via their binding to the FG repeats thereby allowing their passage through the pore (*Hülsmann et al., 2012*; *Frey and Görlich, 2007*; *Frey et al., 2006*). More complex models have been proposed that take into account the sequence inhomogeneity and local molecular properties of the FG nups, their possible spatial localization and dynamics (*Kim et al., 2013*; *Patel et al., 2007*; *Yamada et al., 2010*; *Peters, 2009*; *Mincer and Simon, 2011*; *Cardarelli et al., 2012*; *Ma et al., 2012*; *Solmaz et al., 2013*; *Lowe et al., 2015*). It is likely that the majority of the effects invoked in all these models contribute to the NPC transport mechanism to

some degree. In particular, the FG nups possess various degrees of intra- and inter-chain 'cohesiveness' that can lead to formation of single and multi-chain aggregates (*Schmidt and Görlich, 2015*; *Frey et al., 2006*; *Patel et al., 2007*; *Yamada et al., 2010*; *Hough et al., 2015*; *Milles and Lemke, 2011*).One major contribution to FG nup cohesiveness is believed to arise from the weak binding of the hydrophobic FG repeats to each other. However, intrinsically disordered proteins are notoriously prone to aggregation and the cohesiveness can have multiple sources, including electrostatic, $\pi - \pi$ and $\pi$-charge interactions as well as non-specific interactions between the non-FG parts of the chains (*Uversky, 2002*; *Song et al., 2013*; *Borg et al., 2007*; *Milles et al., 2013*). Cohesiveness of individual FG nups correlates with the ratio of the numbers of the hydrophobic to charged residues in their sequence (*Yamada et al., 2010*). However, the relative importance of the cohesiveness and its specific role in the transport mechanism are still under debate.

To add to the complexity of the system, the transport proteins are present in large numbers within the NPC, and can strongly affect FG nup conformations and dynamics (*Zilman et al., 2007*; *2010*; *Zilman et al., 2009*; *Jovanovic-Talisman et al., 2009*; *Yang and Musser, 2006*; *Lowe et al., 2015* ; *Kapinos et al., 2014*; *Schoch et al., 2012*; *Milles et al., 2013*). However, several controversies persist with respect to their contributions to the architecture and the function of the NPC. Specifically, it is still under debate whether the transport proteins swell or compact assemblies of FG nups (*Kapinos et al., 2014*; *Wagner et al., 2015*; *Eisele et al., 2010*; *2013*).

Understanding the spatial organization and the collective dynamics of the FG nups and the transport proteins during the translocation processes is hindered by the scarcity of experimental methods and insufficient measurement accuracy to assess them in vivo on the relevant time (several milliseconds) and length (several nanometers) scales (*Cardarelli et al., 2012*; *Ma et al., 2012*; *Yang and Musser, 2006*; *Yang et al., 2004*; *Grünwald et al., 2011*; *Dange et al., 2008*). Consequently, computational and theoretical approaches - from atomistic to highly coarse grained - have become increasingly important in the investigations of the possible configurations of the FG nups and the transport dynamics within the NPC (*Zilman et al., 2007*; *2010*; *Mincer and Simon, 2011*; *Opferman et al., 2013*; *Osmanović et al., 2013a*; *2013b*; *Tagliazucchi et al., 2013*; *Moussavi-Baygi et al., 2011a*; *2011b*; *Ghavami et al., 2014*; *Ando et al., 2014*; *Gamini et al., 2014*). However, parameterizations of such models still remain difficult because of the sheer complexity and diversity of the FG nups, exacerbated by conflicting or non-existent measurements of the relevant parameters. The existing parameterizations differ significantly in their physical assumptions and outcomes (*Tagliazucchi et al., 2013*; *Ghavami et al., 2014*; *Ando et al., 2014*; *Gamini et al., 2014*). As a case in point, even the radii of gyration of the FG nups and their interaction affinities with the transport proteins are still under debate (*Yamada et al., 2010*; *Kapinos et al., 2014*; *Tetenbaum-Novatt and Rout, 2010*; *Isgro and Schulten, 2005*; *Eisele et al., 2010*).

For further progress, it is imperative to establish the most pertinent physical features and variables controlling the FG nup conformations induced by the transport proteins. Recent in vitro studies provide the basis for systematic understanding of the conformations of the assemblies of the FG nups with transport proteins in order to relate their molecular properties to their nanomechanical behavior (*Hülsmann et al., 2012*; *Frey and Görlich, 2007*; *Schmidt and Görlich, 2015*; *Lim et al., 2007*; *Frey et al., 2006*; *Kapinos et al., 2014*; *Wagner et al., 2015*; *Eisele et al., 2010*; *2013*). In one typical experimental setup, an FG nup assembly is grafted to a rigid surface in the presence of varying concentrations of the transport proteins (see *Figure 1*). The FG nup conformations are inferred from the measurements of layer height. Even this relatively simple experimental setup resulted in conflicting observations. Depending on the experimental conditions and the measurement technique, the transport proteins can either increase or decrease the layer height, or in some cases cause no measurable change. However, despite the variability between different FG nups and transport proteins, general behavior motif is emerging from these experiments. Typically, at low concentrations, the transport proteins do not affect the layer height. This is followed by a partial layer collapse and height decrease, accompanied by accumulation of the transport protein in the layer. Increasing the concentration further reverses the collapse and eventually leads to the swelling of the layer. Virtually all studied mixtures of the FG nups and transport proteins regardless of the species of origin or the natural localization in the NPC exhibit this general pattern of behavior although the degree of collapse and swelling may vary (*Lim et al., 2006*; *2007*; *Kapinos et al., 2014*; *Wagner et al., 2015*; *Eisele et al., 2010*; *2013*). Related patterns of behavior are observed in experiments with bulk solutions of FG nups mixed with various transport proteins. At sufficiently

high concentrations, the FG nups form a dense phase which either absorbs or excludes the transport proteins, depending on the size of the latter and their interaction strength with the FG nups see *Figure 1*. The behavior is very general and is observed in a wide range or FG nups from different species (*Hülsmann et al., 2012*; *Frey and Görlich, 2007*; *Schmidt and Görlich, 2015*; *Hough et al., 2015*).

The generality of these behaviors - despite the large number of molecular factors affecting the FG nup behavior - suggests that it might be understood in terms of a small number of core organizing principles. In this paper we develop a coarse grained theory of the transport proteins-FG nup assemblies that captures only the essential physical features: (i) the flexible nature of the FG nups, (ii) their potential cohesiveness and (iii) the attractive interactions with the transport proteins. The model is investigated using mean field theory supported by coarse-grained simulations (*Opferman et al., 2012*; *Opferman et al., 2013*). Our approach is inspired by the successes of the simplified theories in explaining the properties of highly chemically complex and diverse materials in polymer science and soft condensed matter (*Doi and Edwards, 1998*; *Flory, 1953*; *de Gennes, 1979*). Systematic comparison of the model predictions with extensive experimental data shows that the model captures and explains the observed behavior in different regimes. The model suggests a resolution of some of the apparent conflicts in the experimental results and proposes how to reconcile the outstanding controversies regarding the relative importance of the cohesive and the entropic effects for FG nup behavior and NPC selectivity. The model also sheds light on the long standing discrepancies in the measurements of the binding affinities of transport proteins to FG nups.

The model identifies the key physical variables controlling the conformational behavior of FG nup-transport protein assemblies and suggests experimental ways of manipulating them. This essential theoretical framework can be systematically developed in the future with additional molecular and structural details. Beyond the NPC, the results of the model are interesting in a broader context of intrinsically disordered proteins and nanotechnological applications (*Stuart et al., 2010*; *Tagliazucchi and Szleifer, 2015*; *Coalson et al., 2015*). Many aspects of the unfolded protein behavior are still puzzling, and the conceptual and computational frameworks - many of which are built on the foundation of polymer physics - are currently being developed (*Uversky, 2002*; *van der Lee et al., 2014*; *Tcherkasskaya et al., 2003*; *Das et al., 2015*; *Sherman and Haran, 2006*). The model described here demonstrates the power of such approaches on a concrete example of an important family of intrinsically disordered proteins.

## Mathematical model

Many of the molecular parameters determining the behavior of the FG nup assemblies and their interactions with transport proteins are not fully known. In particular, the number of binding sites on the transport proteins and affinities of their binding to the FG repeats are a matter controversy. The degree of inter- and intra- FG nup cohesion or the relative importance of the electrostatic vs hydrophobic effects are unknown (*Bayliss et al., 1999*; *2000*; *Frey et al., 2006*; *Yamada et al., 2010*; *Kapinos et al., 2014*; *Schoch et al., 2012*; *Tagliazucchi et al., 2013*; *Tetenbaum-Novatt and Rout, 2010*; *Isgro and Schulten, 2005*; *Eisele et al., 2010*; *Pyhtila and Rexach, 2003*; *Otsuka et al., 2008*). To capture the most salient mechanical and energetic properties responsible for the collective conformational behavior of the FG nups, we model them as flexible polymeric chains. Although not all intrinsically disordered proteins can be described in this way (for instance, due to residual secondary structure), the nanomechanical and the statistical properties of the FG nups can be well described by the polymer worm-like chain model (*Lim et al., 2007*). The chains can be imagined as consisting of 'monomers', where each monomer roughly corresponds to one or several amino-acids; as will be shown below, the predictions of the model turn out to be insensitive to the exact choice of the monomer size within a realistic range (see the Results section). The transport proteins are modeled as rigid particles of appropriate volume that interact attractively with the FG nup chains. Although the interaction between the transport proteins and FG nups is believed to be predominantly mediated by Phenylalanines of the FG repeats lodging themselves into the hydrophobic grooves on the transport proteins, it is likely that other amino acids and other interactions, such as electrostatic interaction, participate in the binding as well (*Bayliss et al., 2000*; *2002*; *Tagliazucchi et al., 2013*; *Isgro and Schulten, 2005*; *Colwell et al., 2010*; *Liu and Stewart, 2005*). Thus, the interactions of the FG nups with the transport proteins are taken into account on a

coarse-grained level parameterized by an interaction parameter $\chi$, which is proportional to the average binding energy of a monomer to a transport protein molecule. The main contributions to $\chi$ are enthalpic, although water network re-arrangement entropy contributes to the hydrophobic interaction as well. We discuss the range of experimentally motivated values of $\chi$ below. The inter- and intra-FG nup cohesive interactions are incorporated into the model in a similarly general fashion through the effective interaction parameter $\chi_{cr}$, as explained below in *Equation (2)*. This description is appropriate for high salt concentrations conditions (150–300 mM) of the experiments, where the electrostatic interactions are highly screened (*Zhulina et al., 2000*; *Barrat and Joanny, 1997*). Similar types of coarse-grained models have been successfully used to describe microtubule-associated unfolded proteins (*Leermakers et al., 2010*), other bio-polymers (*Attili et al., 2012*; *Akinshina et al., 2013*) and DNA-protein interactions (*Jung et al., 2012*). It was shown in *Yamada et al. (2010)* that the observed cohesiveness of different FG nup segments correlates with their hydrophobic to charged amino acid content. Our parameter $\chi_{cr}$ can be thought of as a quantification of this concept and its extension into multi-chain and multi-protein domain.

In addition to the interactions of the FG nups among themselves and with the transport proteins, the main physical factors responsible for the conformations of their assemblies with the transport proteins are the entropy of chain stretching and compression, and the volume constraints. Roughly speaking, the attractive interactions favor more compact structures because they allow formation of higher numbers of energetically favorable contacts, while the entropy of chain configuration favors more diffuse and open conformations. These physical considerations are encapsulated in the free energy that describes an assembly of polymer chains, each of contour length $L$ (*Opferman et al., 2013*; *Milner et al., 1988*; *Halperin et al., 2011*; *Fredrickson et al., 2002*):

$$F = \frac{kT}{2b} \sum_i \int_0^L ds \left( \frac{\partial \vec{r}_i(s)}{\partial s} \right)^2 + \frac{kT}{v_0} \int d^3\vec{r} f\left( \psi(\vec{r}), \phi(\vec{r}) \right). \tag{1}$$

The first term in this free energy stands for the entropic elasticity of the polymer chains; $\vec{r}_i(s)$ describes the trajectory of the $i$-th chain in space and where $b$ is the Kuhn length of the chain, roughly corresponding to the bond length between adjacent 'monomers'. In the second term, $f(\psi, \phi)$ is the local free energy density that includes all the interactions between the polymers and the transport proteins, written in the 'mean field' (well mixed) approximation in terms of their local volume fractions $\psi(\vec{r})$ and $\phi(\vec{r})$, respectively (*Opferman et al., 2012*; *2013*; *Flory, 1953*; *de Gennes, 1979*; *Lai and Halperin, 1992*):

$$f(\psi, \phi) = \frac{1}{\bar{v}} \phi \ln\phi + (1 - \psi - \phi)\ln(1 - \psi - \phi) + \left( \frac{1}{\bar{v}} - 1 \right)(1 - \phi)\ln(1 - \phi) + \frac{1}{\bar{v}}\chi\psi\phi + \frac{1}{2}\chi_{cr}\psi^2, \tag{2}$$

where $v_0 \equiv l^3$ is the monomer volume and $\bar{v} = v/v_0$ is the ratio of the transport protein and monomer volumes (note that a monomer does not have to be spherical). The first term describes the translational entropy of the transport proteins in the layer. The next two terms describe the excluded volume interaction (steric repulsion) between all the monomers of the chains and the transport proteins. The last two terms describe the attractive interaction between the FG nup chains and the transport proteins and among the FG nups themselves, respectively; negative $\chi$ and $\chi_{cr}$ correspond to attraction. Other expressions for free energy can be used and provide qualitatively similar results (*Osmanović et al., 2013a*; *Osmanovic et al., 2012*).

## Surface layer geometry

To further simplify the discussion and establish the key variables controlling the experimentally observed behaviors, we assume that the monomer density is uniform throughout the layer. In reality, the monomer density inside the layer decays away from the grafting surface. We also assume that the entropic elasticity of the chains is described by the Gaussian model (*Alexander, 1977*). This simple approximation cannot be used to predict the exact layer height, but it is a qualitatively good treatment for the moderately cohesive chains and moderate concentrations of the transport proteins of interest in this paper, because the cohesiveness causes layer compaction, as shown below (*Opferman et al., 2012*; *2013*; *Halperin et al., 2011*; *Alexander, 1977*; *Zhulina et al., 1991*; *Lai and Halperin, 1992*). The overall conclusions of this paper do not depend on this approximation (see *Figure 2—figure supplement 1*). The discussion is further simplified by normalizing the

average layer height $h$ by the chain length $L$, introducing a new variable $\overline{h} = h/L$. With these, the free energy per unit area of a layer of chains grafted at a distance $a$ from each other becomes (*Alexander, 1977*; *de Gennes, 1980*)

$$F(\overline{h}, \psi, \phi)/A = kT \frac{L}{l^3} \left[ \frac{\overline{\sigma}\overline{h}^2}{2} \frac{l}{b} + \overline{h} f(\psi, \phi) \right], \tag{3}$$

where $\overline{\sigma} = (l/a)^2$ is the grafting density of the chains normalized by the average monomer cross-section $l^{-2}$ ($l \equiv v_0^{1/3}$). The layer height related to the monomer density through the condition $\overline{\sigma}\frac{l}{b} = \psi\overline{h}$ which expresses the fact that the total number of the monomers in the layer is constant, $hl^3\psi = N\sigma$. At this level of approximation, the equilibrium layer height is found by the minimization of the free energy over $\overline{h}$ and the transport protein concentration $\phi$ under the constraints that the chemical potential of the transport proteins in the layer and the osmotic pressure in the layer are equal to those in the outside solution of volume fraction $c$. Importantly, because $L$ factorizes out of the free energy expression in *Equation (3)*, the resulting equilibrium values of $\overline{h}$ and $\phi$ are independent of the chain length $L$. The chemical potential and the osmotic pressure of the outside dilute solution - assumed to be ideal - are $\mu_c = k_B T \ln(c)$ and $\pi_c = k_B Tc/v$. This procedure has been described in detail and verified by coarse-grained brownian dynamics simulations in *Opferman et al. (2012)*, *Opferman et al. (2013)*. Finally, it is important to keep in mind that the calculated properties are equilibrium average values. On the molecular scale, the polymers are highly dynamic and their individual conformations fluctuate on the microsecond time scale.

## Bulk solutions

In a bulk solution where the chains are not grafted to a surface but are freely floating in solution, the entropic stretching (first term in *Equation (3)*) is replaced by the translational entropy of the chains, so that the mean field free energy per unit volume is *Doi and Edwards (1998)*, *Flory (1953)*, *de Gennes (1979)*

$$f(\psi, \phi) = \frac{1}{N}\psi\ln\psi + \frac{1}{v}\phi\ln\phi + (1 - \psi - \phi)\ln(1 - \psi - \phi) + \left(\frac{1}{v} - 1\right)(1 - \phi)\ln(1 - \phi) + \frac{1}{v}\chi\psi\phi + \frac{1}{2}\chi_{cr}\psi^2. \tag{4}$$

This free energy becomes unstable for sufficiently large interaction parameters $|\chi|$ or $|\chi_{cr}|$, leading to a phase separation where a dense phase of transport proteins mixed with the FG nups coexists with a very dilute solution. The compositions of the dense and the dilute phases are determined from the equality of the osmotic pressures and the chemical potentials of the transport proteins and the FG nups in the coexisting phases (*de Gennes, 1979*; *Zilman and Safran, 2002*; *Morse, 1969*) see Appendix for details.

## Dimensions of individual FG nups

The model can be used to calculate the dimensions of individual FG nup molecules in solutions. Due to the thermal motion, each flexible chain dynamically samples multiple spatial conformations that on average occupy a volume of size $R$, which can be found through minimization of the free energy $\frac{3R^2}{2N} + \frac{4}{3}\pi R^3 f(\psi)$ over $R$ with the condition $\frac{4}{3}\pi R^3 \psi = N$; the first term represents the entropic elasticity of chain conformations in space, while the second term describes the intra-chain interactions where $f(\psi)$ is the free energy of *Equation (2)* with $\phi = 0$. Slightly different expressions for the free energy can be used, all leading to qualitatively the same results (*de Gennes, 1979*; *Sherman and Haran, 2006*; *Sanchez, 1979*).

As shown below, the overall qualitative predictions of the theory are robust with respect to the choice of model parameters within a physically feasible range. However, quantitative or semi-quantitative comparison with the experimental data requires a specific choice of the molecular parameters $b$ and $v_0 \equiv l^3$. The approximate volume of a transport protein molecule can be calculated from its molecular mass and the average protein density $\rho \simeq 1.2 - 1.5$ g/cm$^3$. For Karyopherin-$\beta$1 (molecular mass $\approx 97 - 103$ kD, depending on the attached tag), $v \simeq 120 - 140$ nm$^3$ and for NTF2, the transport protein specialized for the import of RanGDP into the nucleus (molecular mass $\approx 33$ kD), $v \simeq 35 - 45$ nm$^3$. The estimates of the 'monomer' size are somewhat less well defined. The average distance between two adjacent amino acids on a polypeptide chain is $\approx 0.36 - 0.38$ nm and the side chain size

varies in the range $\sim 0.3 - 0.6$ nm (*Levitt, 1976*; *Zamyatnin, 1972*; *Quillin and Matthews, 2000*). However, their effective size can be modulated by the bound ions present within the Debye screening length (of the order of $\sim 0.5 - 1$ nm at the experimental salt concentrations) or bound denaturant molecules. Thus, for comparison with experiments, the realistic monomer size lies within a range $b \approx 0.4 - 1.6$ nm and its volume $v_0 \approx 0.12 - 1$ nm$^3$; the upper limit corresponds to a 'monomer' composed of about four amino acids. Finally, the exact structure of the dense phase and its maximal molecular packing fraction are unknown (*Frey and Görlich, 2007*; *Schmidt and Görlich, 2015*; *Milles et al., 2013*). For the conversions between the theoretical volume fractions and the experimentally measured concentrations, we have assumed the maximal packing fraction $z = 0.625$ in the dense phase, typical of the random close packing of dense molecular assemblies which normally lies in the range of $z = 0.5 - 0.7$ (*Nolan and Kavanagh, 1992*).

## Results

### Conformations of an FG nup layer in the absence of transport proteins

To establish whether the model captures the basic biophysical characteristics of the FG nup assemblies, we first apply it to the case of an FG nup layer without transport proteins. In this case the transport protein density is $\phi = 0$, and the free energy per area $A$ of *Equations (2) and (3)* is minimized over $\bar{h}$ to obtain the equilibrium average layer height. We emphasize again that once the layer height has been re-scaled by the polymer contour length $L$, the theoretical predictions become independent of $L$. This will be important in the analysis of the experimental data.

The conclusions of the model are summarized in *Figure 2*, which shows that the FG nup layer height $h$ decreases with the grafting distance $a$, because the steric repulsion between the polymers - that maintains the chains being stretched on average - is higher at lower grafting distances. The theory predicts that the layer height is monotonically decreasing with the cohesion strength. This is expected because the cohesiveness favors more compact conformations with more favorable contacts while the entropic elasticity term favors more diffuse conformations. In polymer physics parlance, increasing the cohesiveness is similar to changing the solvent 'quality' from 'good' to 'bad' (*Osmanović et al., 2013a*; *Eisele et al., 2013*; *de Gennes, 1979*; *Milner et al., 1988*; *Halperin et al., 2011*; *Lai and Halperin, 1992*; *Zhulina et al., 1991*; *Peleg et al., 2011*; *Moh et al., 2011*). The model also shows that sufficiently strong cohesion not only decreases the layer height, but shifts the behavior into a qualitatively different regime. The FG nup internal conformation and cohesiveness can be characterized by the scaling exponent $g$ that describes the dependence of the height $h$ on the grafting distance $a$, $h \sim a^{-g}$. For an ideal non-cohesive, purely entropically stabilized polymer brush, $g = 2/3$. Cohesiveness increases the value of the exponent towards $g = 2$, in which regime the layer effectively behaves as a material of constant density with $\psi$ independent of $h$ (*de Gennes, 1979*; *Milner et al., 1988*; *Zhulina et al., 1991*; *Moh et al., 2011*). Nevertheless, even in the highly cohesive regime, the layer has significant 'free space' occupied by the solvent (calculated as $1 - \psi$).

Behavior of the FG nup layers reported in *Kapinos et al. (2014)*, *Schoch et al. (2012)*, *Wagner et al. (2015)* follows the general predictions of the theory. In particular, examination of the data shows that the height of FG nup layers in the absence of transport proteins decays faster than $a^{-2/3}$ with the grafting distance $a$, but slower that $a^{-2}$, as shown in *Figure 3*. The accuracy of the experimental measurements does not allow one to quantitatively differentiate between different FG nups based on the scaling exponent $g$. However, interpreted in light of the theoretical model, the data strongly indicate the presence of significant cohesion in all FG nups. Importantly, Nsp1 segments of different lengths fall within the same family of curves, once normalized by their length, in accord with the theoretical predictions. These results indicate that the salient physical mechanisms responsible for the behavior of grafted FG nup layers are adequately captured by the model.

### Effects of the transport proteins on the layer height

We now compare the theoretical predictions with the experimental data in the presence of transport proteins. Although at a first glance experimentally observed responses of different FG nups to the addition of transport proteins appear rather different (*Kapinos et al., 2014*; *Eisele et al., 2010*), closer inspection shows that most FG nups exhibit the same general pattern of behavior. With

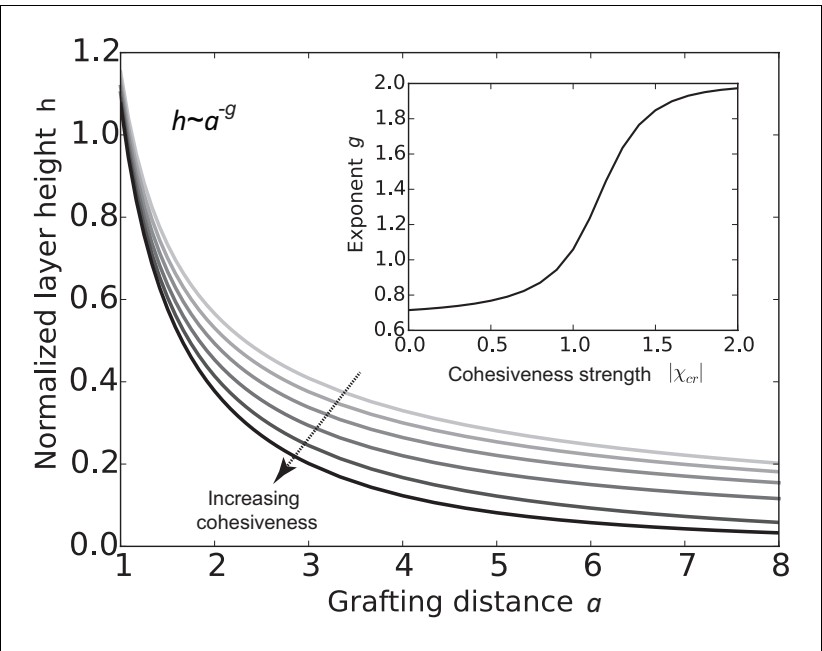

**Figure 2.** Cohesion makes FG nup layers more compact: theoretical predictions. Layer height $h/L$ normalized by the chain length as a function of the normalized grafting distance $a/l$ for increasing cohesiveness ($\chi_{cr}$ varies from $\chi_{cr} = 0$ to $\chi_{cr} = -1.5$). For any value of $\chi_{cr}$, the curve is well approximated by the dependence $h \sim a^{-g}$. The inset shows that the exponent $g$ increases from $2/3$ to $2$ as the absolute value of the cohesion strength $|\chi_{cr}|$.

The following figure supplement is available for figure 2:

**Figure supplement 1.** Effect of density non-uniformity on the model predictions.

progressive addition of the transport proteins, the layer height decreases to some extent followed by recovery and eventual swelling. The initiation of the collapse is correlated with the penetration of the transport proteins into the layer. We have recently shown that this behavior is expected on very general grounds for polymer layers infiltrated with nanoparticles (*Opferman et al., 2012*; *2013*). This common behavior is shown in *Figure 4*, which renders the experimental data from *Kapinos et al. (2014)*, *Wagner et al. (2015)*.

Intuitively, the penetration of an individual transport protein into the layer is determined by the balance between the energetic (enthalpic) gain of creating more contacts with the FG nups and the entropic cost of displacing and crowding the FG nup chains. If the overall free energy change upon insertion of one transport protein into the layer is negative, it will typically penetrate the layer. Otherwise, the penetration is exponentially suppressed, although there still will be some particles in the layer. For a single particle of radius $R$, at low particle concentration, the entropic cost of penetrating an ideal polymer brush layer can be estimated as $\simeq \alpha k T R^2 \psi$, where the prefactor $\alpha$ depends on the grafting density and the degree of cohesiveness (*Halperin et al., 2011*; *Egorov, 2012*; *Milchev et al., 2008*). On the other hand, a rough estimate for the energetic/enthalpic gain contacts is $\simeq -\epsilon n \psi$, where $n$ is the number of the interaction sites on the protein and $\epsilon$ is the energy per contact. Thus, for $\epsilon < \alpha k T n/R^2$ the entropic repulsion dominates, and one does not expect significant penetration into the layer. In principle, this entropic repulsion from the flexible polymer layer is sufficient for creating the permeability barrier for non-binding molecules. It is crucial to bear in mind that the polymer chains are not static but highly fluctuating entities. It is the entropy of these molecular motions that is responsible for the penetration barrier; thinking of a polymer layer as a static entity with some amount of free space can lead to erroneous conclusions. The barrier could be enhanced by other effects, such as, for instance, inter-chain cohesion.

Thus, at low concentrations the transport proteins penetrate the layer if their attractive interaction with the FG nups is strong enough. However, they do not cause significant conformational changes -

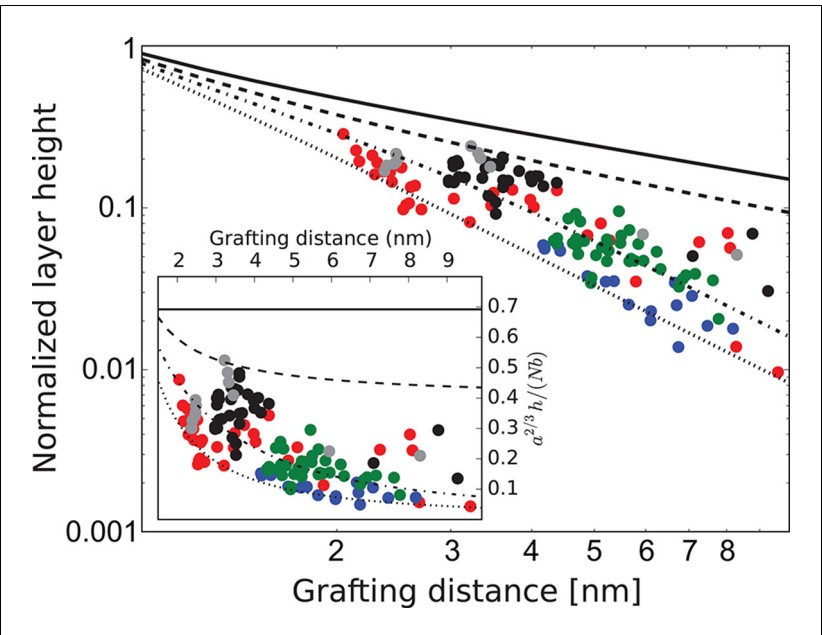

**Figure 3.** FG nup layer height depends on the grafting distance: theory vs. experiment. The dots are the experimentally measured layer heights from *Kapinos et al. (2014)* and *Wagner et al. (2015)* normalized by the FG nup length. Red, blue, green and black: grafted layers of Nup62, Nup98, Nup153 and Nsp1, respectively; gray dots belong to a short Nsp1 segment. Solid line: $h \sim a^{-2/3}$ is the ideal brush ($\chi_{cr} = 0$) behavior obtained from the model. Dotted line: $h \sim a^{-2}$ is the behavior of a strongly collapsed brush with $\chi_{cr} = -2.5$. All the FG Nups lie between these two regimes, indicating a significant amount of cohesion for all FG nups; the dashed line is for $\chi_{cr} = -0.8$; the dashed-dotted line is for $\chi_{cr} = -1.4$. To enhance the contrast, inset shows the same data with the height $h$ normalized by the ideal brush height. $b = 1.52$ nm, $l = 1$ nm.

essentially occupying the available empty space inside the layer. At higher densities of the transport proteins, or higher interaction strengths, the number of transport proteins in the layer increases, and collective effects start to play a role (*Opferman et al., 2013*; *Halperin and Kröger, 2011*; *Kim and O'Shaughnessy, 2006*). Further addition of the transport proteins causes a cooperative conformational transition of the FG nups leading to either collapse or swelling of the layer, correlated with the accumulation of the transport proteins inside the layer. The magnitude of the collapse and swelling depend on the transport protein size, concentration, interaction strength with the FG nups, the grafting density, and the cohesion strength. Typical height responses to transport protein concentration are shown in *Figure 5*. One important conclusion of the theory is that the FG nup response to the addition of the transport proteins (for instance, 'swelling' vs. 'collapse') is not an intrinsic property of an FG nup, but can be modulated by the grafting density, transport protein size and the interaction strength. The overall repertoire of predicted behavior as a function of parameters is shown in the 'phase diagrams' in *Figures 6* and *7*. The model captures the general salient features of the experimental observations. In particular, at the same grafting distance and the cohesion strength, the collapse is more pronounced for a smaller protein such as NTF2, in accord with the experimental observations (*Wagner et al., 2015*). Similarly, higher grafting distance results in more collapse - because the entropic repulsion of the transport proteins by the chains is lower at lower chain density. Different experimental results can be placed in the different parts of the 'phase diagram', potentially explaining the observed discrepancies. Notably, the theory predicts a high degree of collapse at large grafting distances, in agreement with the experiments of (*Lim et al., 2007*). Another interesting prediction of the model is that changing the grafting distance affects small and large transport proteins in a different way. For large ones (e.g. Karyopherin), increasing the grafting distance enhances the penetration into the layer because it reduces the repulsive barrier due to lower monomer density in the layer. By contrast, for small transport factors (such as NTF2), for which

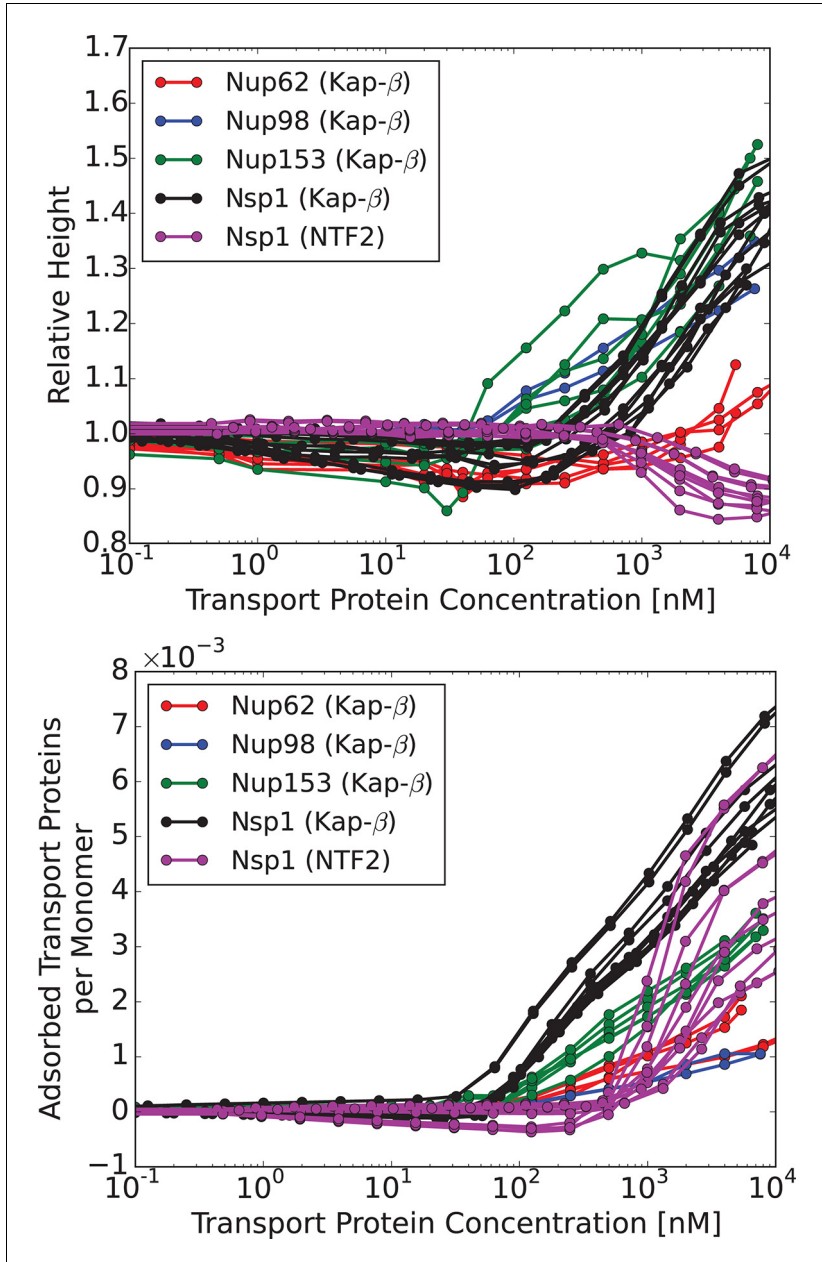

**Figure 4.** Characteristic responses of FG nup layers to the transport proteins: experimental results. *Upper panel*: change in the layer height relative to the unperturbed layer as a function of the transport protein concentration in the outside solution. *Lower panel*: number of the transport proteins in the layer per unit length of the FG nup chain. Each line corresponds to a different run with a different initial layer height and grafting distance. Different colors correspond to different FG nups, which all exhibit qualitatively similar behavior. *Color coding.* Red, blue, green and black: Karyopherin-$\beta$1 interacting with Nup62, Nup98, Nup153 and Nsp1, respectively; magenta: NTF2 interacting with Nsp1. The corresponding average grafting distances are $\sim 2.5$ nm, $\sim 4$ nm, $\sim 4.5$ nm, $\sim 3.7$ nm. The data are from Refs. (*Kapinos et al., 2014*; *Wagner et al., 2015*)

The following figure supplement is available for figure 4:

**Figure supplement 1.** Nup214.

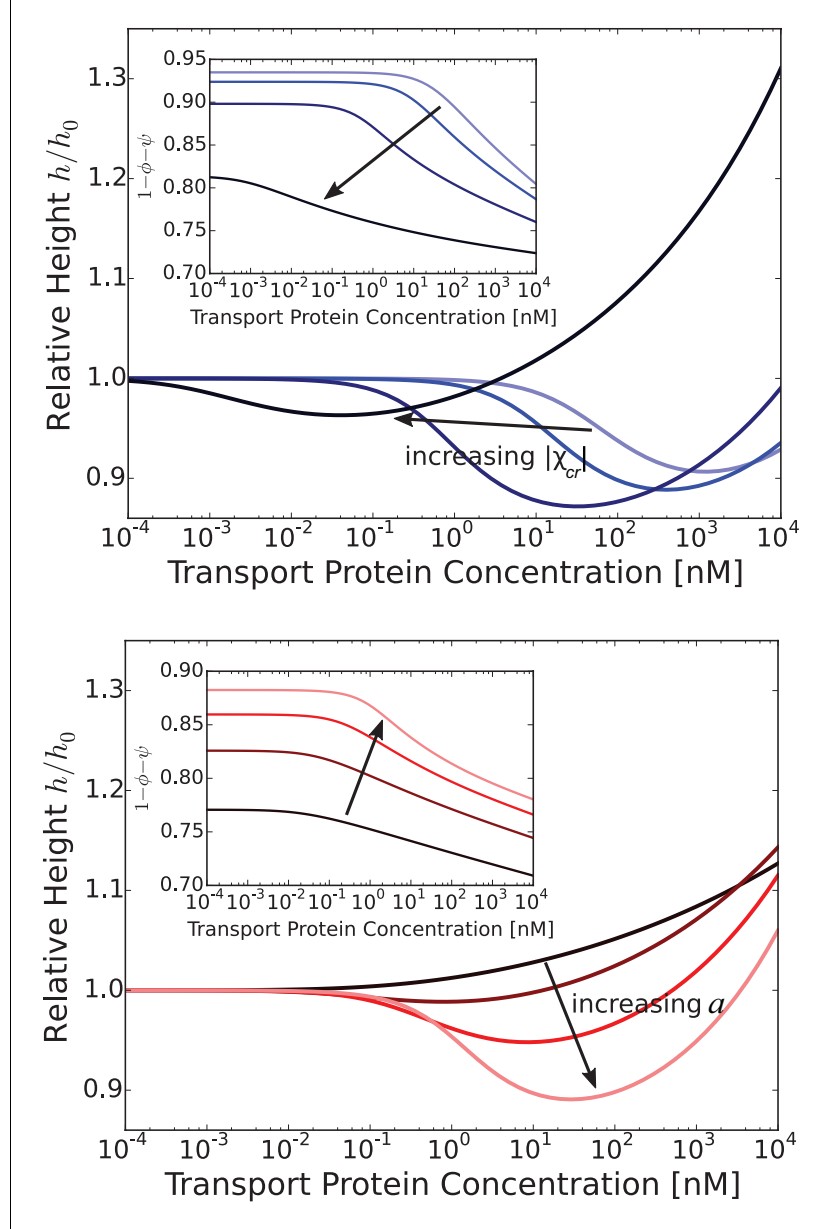

**Figure 5.** Layer collapse and swelling: effect of cohesion and of the grafting distance. *Upper panel:* Theoretical curves show that FG nup cohesion can convert layer collapse to swelling. The cohesion strengths are $\chi_{cr} = 0, -0.4, -0.8, -1.1$ for $a = 5$ nm and $\chi = -550$. *Lower panel:* Increasing grafting distance increases the magnitude of the layer compaction. The lines correspond to model predictions for $a = 3, 4, 5, 6$ nm for $\chi = -530$ and $\chi_{cr} = -1$. The insets show that the fraction of free space in the layer, calculated as $1 - \phi - \psi$, decreases with the addition of the transport proteins. $b = 1$ nm, $l = 0.67$ nm in both panels.

the repulsion is less important, increasing the grafting distance decreases the penetration into the layer because it reduced the density of available binding sites in the layer.

Can this simplified theory be qualitatively or semi-quantitatively related to the experimental data for realistic values of parameters? Due to the experimental accuracy limitations and large uncertainly in the known values of all the parameters, the fitting of the parameters to the data is not unique and has limited information content. Rather, we focus on clearly distinguishable trends, such as the difference between the behavior of Karyopherin-$\beta$1 vs. NTF2 on the Nsp1 layer, reported in *Wagner et al. (2015)*. These measurements, performed on the same FG nup in approximately same

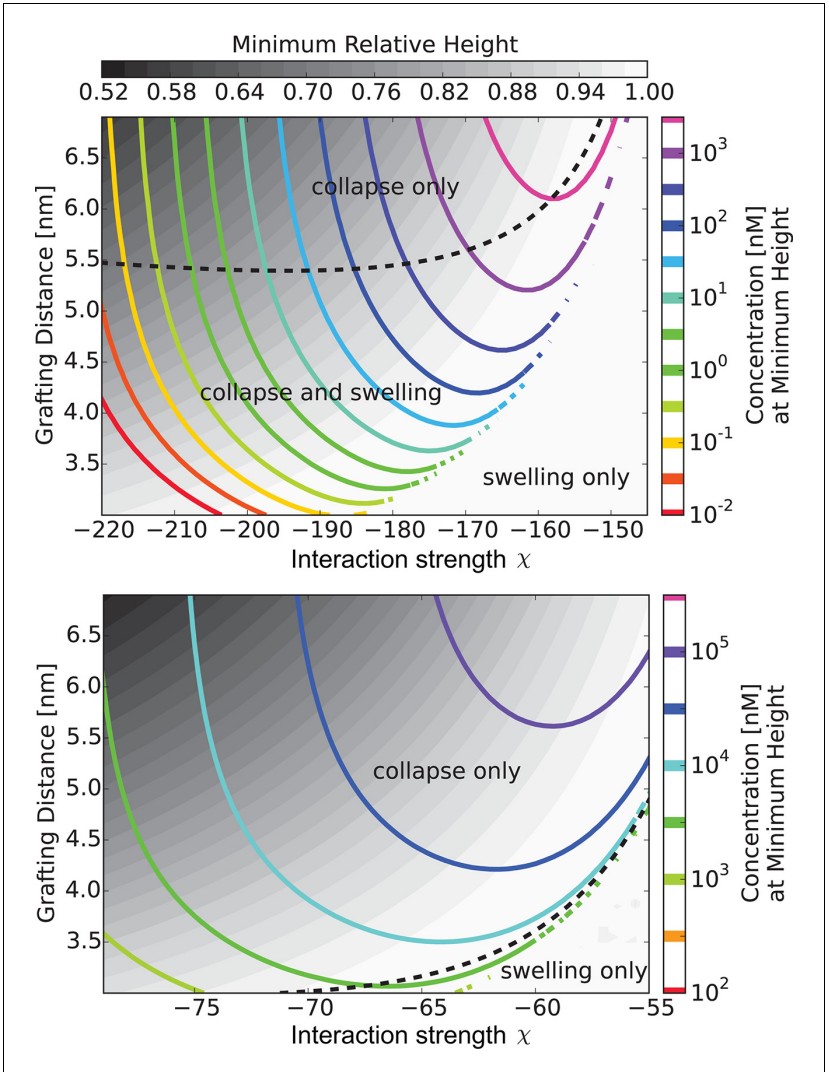

**Figure 6.** The 'phase diagram' of predicted behaviors: conformational transitions of the layer. The grayscale color denotes the degree of layer compaction, $h_{min}/h_0$, relative to the unperturbed layer (color legend is on top). The colored contour lines indicate the corresponding bulk concentration $c_{min}$ at which the minimal layer height is achieved (legend on the right side). There is no layer swelling above the dashed line (up to 1 $\mu$M transport protein concentration). *Upper panel*: $\bar{v} = 125$, roughly corresponding to Karyopherin-$\beta$1; *Lower panel*: $\bar{v} = 40$, roughly corresponding to NTF2. The overall phase diagram topology is similar in both cases, but for smaller protein the collapse is more pronounced and occurs at lower interaction strengths $\chi$. In both panels $b = 1.52$, $l = 1$ nm, corresponding to the 'monomer' size of roughly four amino acids.

The following figure supplement is available for figure 6:

**Figure supplement 1.** Model predictions are robust with respect to the monomer size estimate.

---

range of grafting distances, allow us to examine the effects of the transport protein size and the binding strength, unconfounded by other factors. Karyopherin-$\beta$1 and NTF2 have significantly different sizes and binding strengths. NTF2 is ~ 4 times smaller in volume and has only two binding sites, while Karyopherin-$\beta$1 can have up to ten specific FG-binding sites and a significantly larger surface area with potentially much larger number of non-FG interactions with the FG nups (***Bayliss et al., 1999***; ***2000***; ***2002***; ***Isgro and Schulten, 2005***; ***Liu and Stewart, 2005***). We focus on one clearly discernible difference in the experimental behavior: significant penetration of NTF2 starts at higher concentrations but causes stronger compaction of the layer compared to Karyopherin-$\beta$1. Comparison

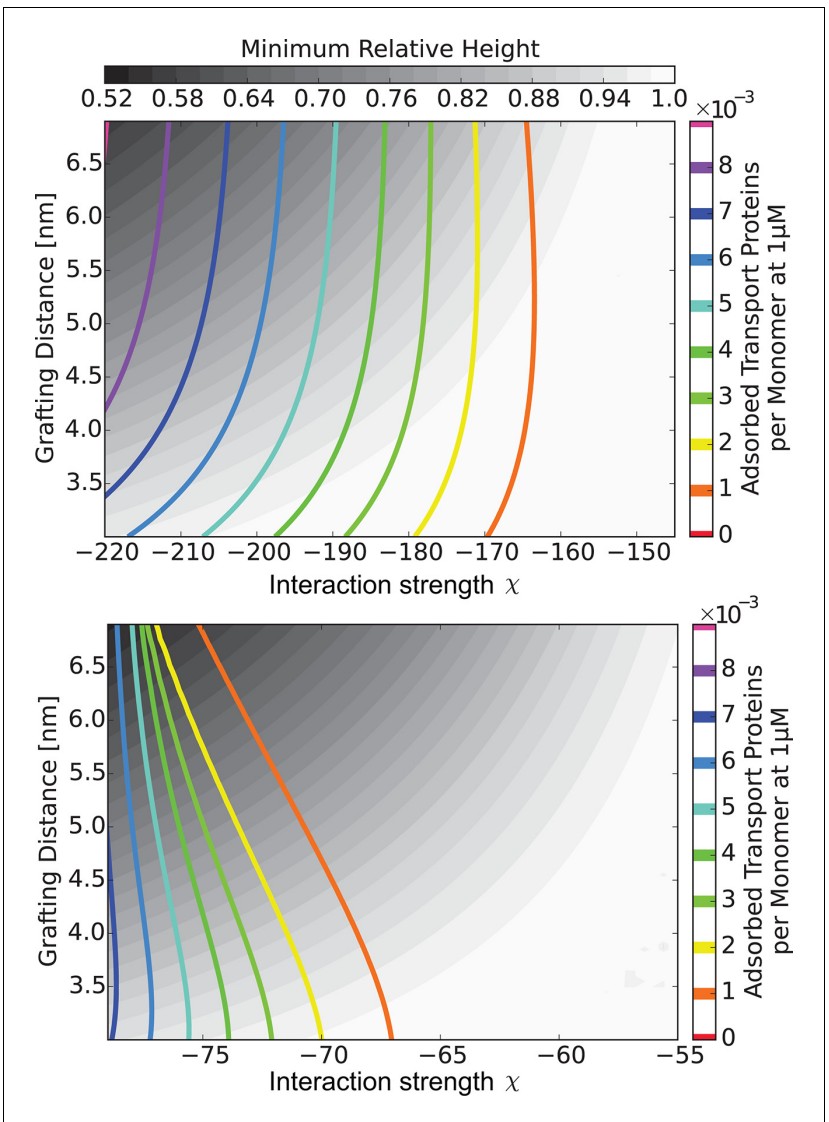

**Figure 7.** The 'phase diagram' of predicted behaviors: amount of transport protein in the layer. The grayscale color denotes the degree of layer compaction, $h_{min}/h_0$, relative to the unperturbed layer (color legend is on top). The colored contour lines show the amount of adsorbed proteins in the layer per chain monomer. Higher degree of collapse is correlated with higher accumulation of the proteins in the layer. *Upper panel*: $\bar{v} = 125$, roughly corresponding to Karyopherin-$\beta$1; *Lower panel*: $\bar{v} = 40$, roughly corresponding to NTF2. In both panels $b = 1.52$, $l = 1$ nm.

of the theoretical predictions with the experimental results is shown in *Figure 8* in the range of concentrations where direct comparison is possible. Because of the relatively large uncertainty in the measurements of the absolute layer height, the grafting distance (see *Figure 4*) and the binding strengths, the theoretical predications are shown for a range of values approximately corresponding to the experimental ones. The model reproduces the observed differences in the behavior of Karyopherin-$\beta$1 and NTF2 on Nsp1 at physically plausible values of the parameters in the regime of its validity. It might also explain why no significant change in the layer height (or very limited swelling) was observed upon addition of transport proteins by other experimental groups (*Eisele et al., 2010; 2012*): the behavior of both NTF2 and Karyopherin-$\beta$1 can be easily shifted into the swelling regime by relatively small changes in the grafting distance, cohesion or interaction strength, the latter of which can be modulated by small changes in the *pH* or salt and denaturant concentrations. Other patterns revealed by the experimental data shown in *Figure 4*, such as the dependence of the

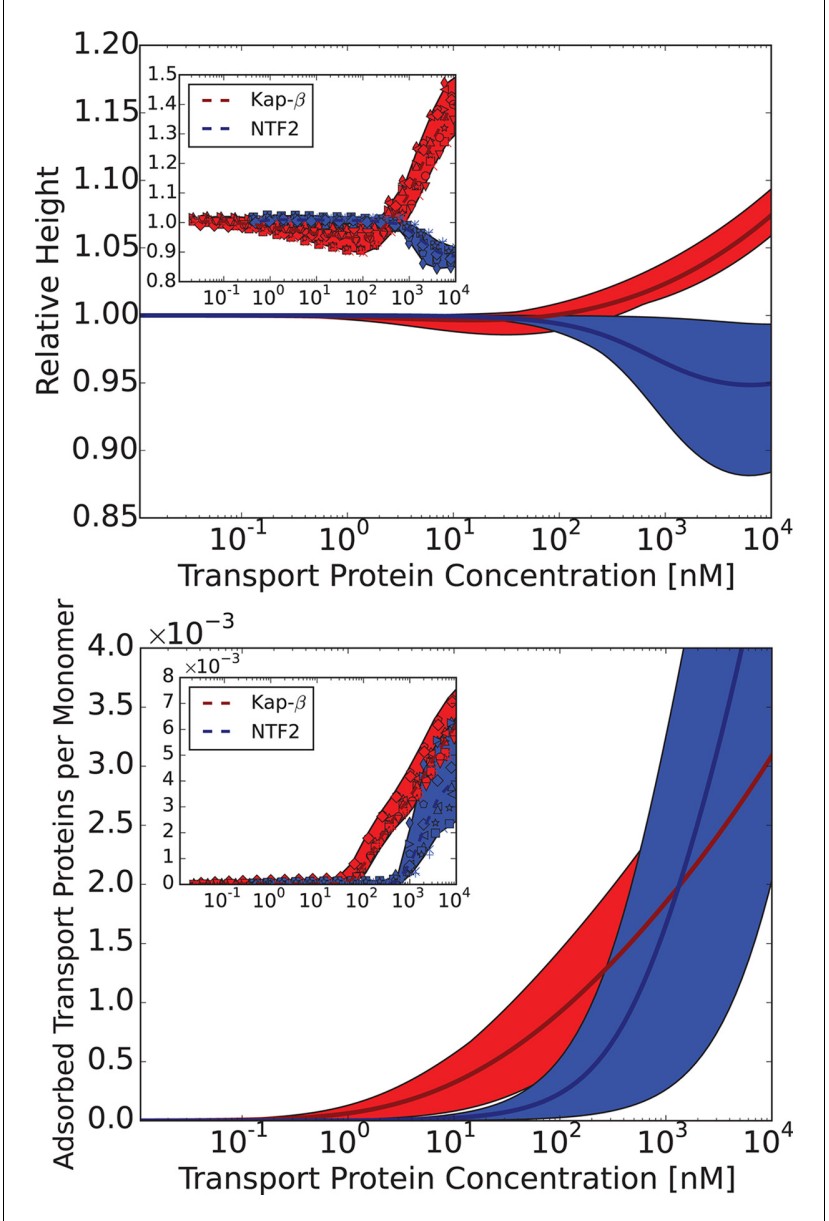

**Figure 8.** Comparison of the theoretical predictions with the experimental data in the layer geometry. Theoretical predictions for the range of the parameter values approximately corresponding to the experimental ones for Nsp1 layers infiltrated by Karyopherin-$\beta$1 and NTF2. *Upper panel*: layer height vs bulk concentration of the transport protein. Red: Karyopherin-$\beta$1, Blue: NTF2. *Lower panel*: amount of adsorbed transport protein in the layer as a function of the concentration in the solution. Red: Karyopherin-$\beta$1, Blue: NTF2. The shaded regions correspond to $3.5 < a < 4$ nm and $-73 < \chi < -63$ for NTF2 and $-185 < \chi < -175$ for Karyopherin-$\beta$1. For all lines, $b = 1.52$, $l = 1$ nm and $\chi_{cr} = -1$. The insets show the corresponding experimental data from *Wagner et al. (2015)*.

The following figure supplement is available for figure 8:

**Figure supplement 1.** Model predictions are robust with respect to the monomer size estimate.

maximal compaction concentration on the grafting distance, are also qualitatively explained by the theory. Quantitative comparison of these features requires more analysis of the data and more detailed approximations, including the sparse and high transport protein density regimes, and will be presented elsewhere.

| TP-cargo complex | Schematic representation | MW(kDa) | Results |
|---|---|---|---|
| NTF2 | | 33 | TP-cargo penetrate into Nup aggregates |
| Imp β - IBB | | 103 | TP-cargo penetrate into Nup aggregates |
| Imp β - IBB-GFP | | 131 | No penetration |
| Imp β - IBB-MBP-GFP | | 172 | No penetration |
| (Imp β - ZsGreen) × 4 | | 520 | TP-cargo penetrate into Nup aggregates |

**Figure 9.** Partitioning of transport proteins into dense FG nup phase: summary of experimental results. Experimentally, partitioning of the transport proteins (TP) complexes with various cargoes into the dense FG nup phase depends on the cargo size and the overall interaction strength of the complex with the FG nups. Both Importin-$\beta$ (vertebrate homologue of Kap-$\beta$1) and NTF2 penetrate the dense phase, but the Importin-$\beta$ with either medium (IBB-GFP) or large (IBB-MBP-GFP) cargo does not. However, the very large complex of four Importin-$\beta$ complexed with four ZsGreen proteins partitions into the dense phase. The results are for the dense phase of TtNup98 of *Tetrahymena Thermophila* recombinantly expressed in bacteria, adapted from Schmidt and Görlich (*Schmidt and Görlich, 2015*).

The chosen values of $b$ and $l$ correspond to the 'monomer' size of approximately four amino acids - roughly the size of one FG patch, reflecting the fact that FG nup interaction with the transport proteins requires this particular local sequence of amino acids (*Bayliss et al., 2000*; *2002*); we emphasize that our model does not correspond to a randomized amino acid sequence of the FG nups (*Tagliazucchi et al., 2013*; *Ghavami et al., 2014*). The interaction parameter roughly $\chi$ corresponds to the attractive part of the second virial coefficient of the interaction between the transport proteins and the monomers, so that $\chi \simeq n \frac{\epsilon}{kT} e^{\epsilon/kT}$, where $n$ is the number of available binding sites on the protein, and $\epsilon$ is the average binding energy of one site (*Doi and Edwards, 1998*; *Pathria, 1996*). Taking into account the possible non-FG interactions, $n \sim 6 - 14$ for Karyopherin and $n \sim 2 - 5$ for NTF2. This translates to $\epsilon \sim 2.5 - 3kT$ for the chosen parameter values, which is a reasonable estimate for the average energies of the weak hydrophobic and electrostatic interactions in question. These numbers also agree with recent estimates using other methods (*Kapinos et al., 2014*; *Tu et al., 2013*). Similarly, assuming that the main contribution to cohesiveness comes from the FG-FG interactions, the value of $\chi_{cr} = -1$ corresponds to $\epsilon_{cr} \simeq 2kT$. These values will guide our analysis of bulk solutions of FG nups and transport proteins in the next section. Importantly, the general agreement between the theory and experiment is robust with respect to parameterization choices (see *Figure 6—figure supplement 1*).

The model also sheds light on another long-standing controversy in the field - the high variability among different experimental groups of the measured affinities of the transport protein binding to the FG nups, with values of the measured dissociation constant ranging from several nanomolars to several micromolars (*Bayliss et al., 1999*; *Kapinos et al., 2014*; *Eisele et al., 2010*; *Pyhtila and Rexach, 2003*; *Tetenbaum-Novatt et al., 2012*). As can be seen in *Figure 8*, due to the collective effects and the conformational changes during binding, the adsorption curves are not well described by the standard Langmiur curve typically used to quantify the interaction in binding assays

(*Kapinos et al., 2014*). An attempt to fit it with one or a combination of Langmiur isotherms would lead to different results depending on the concentration range and the grafting density (*Schmidt and Görlich, 2015*). The model developed here provides fundamental physical reasons for the discrepancies in experimental measurements.

## Phase separation in bulk solutions of FG nups mixed with transport proteins

The fundamental physical considerations underlying the behavior of the surface assemblies of FG nups manifest themselves also in the behavior of mixtures of the FG nups and the transport proteins in bulk solutions. Such mixtures were systematically studied in *Schmidt and Görlich (2015)* over a wide range of FG nups from different species (recombinantly expressed in bacteria), systematically varying the size of transport protein-cargo complexes. It was found that even in the absence of transport proteins, solutions of Nup98 FG nucleoporin phase separate into a dense phase with a very high protein concentration, in equilibrium with a very dilute solution. Importantly, unlike the previously reported 'gels' of Nup98 and other FG nucleoporins (*Hülsmann et al., 2012*; *Frey and Görlich, 2007*), these phases form via an equilibrium phase separation mechanism, allowing comparison with our model. Upon addition of cargo-carrying transport proteins bound to such FG nup solutions, the transport protein-cargo complexes either penetrate the dense phase, or stay predominantly in solution, depending on their size and the interaction strength with the FG nups. These results are summarized in *Figure 9*. In this section, we show that the observed patterns naturally follow from the minimal model of this paper.

### Binary FG nup/buffer solutions

Phase separations of polymer solutions with significantly strong inter-chain cohesiveness are well understood. Phase separation is expected when the cohesiveness is strong enough to make the formation of the dense phase energetically favorable to overcome the loss of the chain entropy in the compact aggregate, $|\chi_{cr}| > 1 + \frac{2}{N^{1/2}}$ (in the mean field approximation for $N \gg 1$); note that this condition is independent of any other parameters such as the monomer volume $v_0$ (*de Gennes, 1979*). FG nup densities in the co-existing phases can be found from the equality of the FG nup chemical potentials and the osmotic pressures, as explained in the Methods and the Appendix.

The model predictions for the phase separation in FG nup solutions agree very well with the experimental data, as shown in *Figure 10*. The comparison is made for the for the same values of the parameters $b$ and $v_0$ as in *Figure 3* that describes the surface layers FG nups. The inferred value of the cohesiveness strength $\chi_{cr} \approx -1.5$ for Nup98, known to be more cohesive than Nsp1, is also consistent with the behavior of the grafted layers shown in *Figure 3*. Taken together, this indicates that the model captures the essential biophysical properties of the FG nup assemblies both in surface grafted geometry and in the bulk solutions.

The model also captures biophysical properties of different FG nups reported in *Yamada et al. (2010)*, where dimensions of individual FG nups were shown to correlate with their hydrophobic to charged amino acid content ratio. The differences between the cohesive collapsed coils - such as Nsp1n (N-terminal domain of Nsp1), Nup100n, Nup116m - and the non-cohesive extended coils, such as Nsp1m (C-terminal domain of Nsp1), are captured by different values of $\chi_{cr}$, as shown in *Figure 11* and *Figure 11—figure supplement 1*, in accord with the interpretation of Yamada et al (*Yamada et al., 2010*). Importantly, the values of $\chi_{cr}$ arising from the analysis of the data from *Yamada et al. (2010)* are entirely consistent with those obtained from the analysis of the surface assemblies of Nsp1 in experiments by Lim and coworkers (*Kapinos et al., 2014*; *Wagner et al., 2015*) (which used an overlapping segment of the C-terminal domain) and of bulk assemblies of Nup98, Nup110 and 116 in the work by Gorlich and collaborators (*Schmidt and Görlich, 2015*). Transition from the 'extended coil' regime to the 'collapsed coil' regime upon increase in the chain cohesiveness strength is known in polymer physics as the 'coil-globule' transition and stems from the same physical factors that govern the changes in height of the grafted layers described in *Figure 2*.

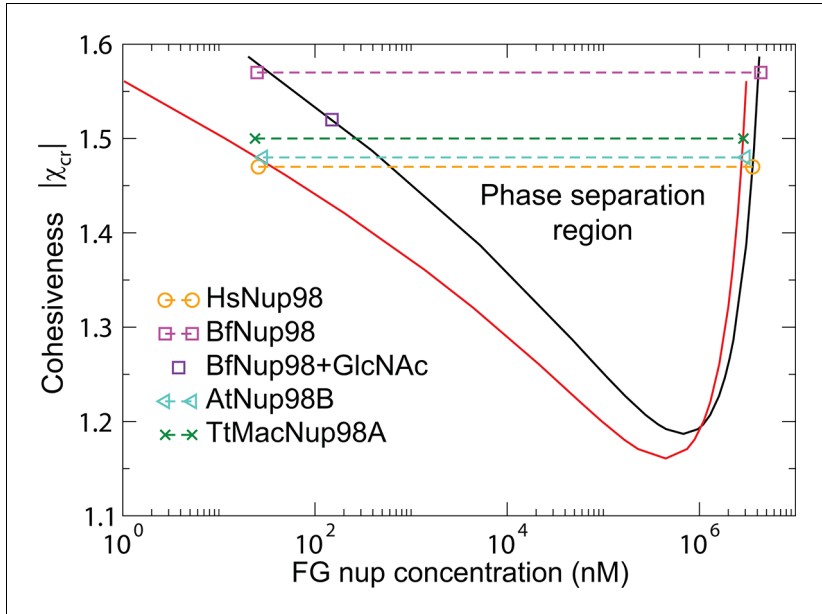

**Figure 10.** Phase separation in FG nucleoporin solutions: theory vs. experiment. For combinations of $\chi_{cr}$ and FG nup concentrations that lie above the phase separation boundaries shown in solid lines, the system undergoes a phase separation into the dilute and the dense phases that lie on the boundaries of the phase separation region. The symbols show the experimentally estimated concentrations of Nup98 from different species in the co-existing dilute and dense phases from **Schmidt and Görlich (2015)**. The phase separation boundaries are calculated with the following parameters: N=125 monomers (black) and N=167 monomers in the chain (red), $l = 1$ nm$^3$, $b = 1.52$ nm, corresponding to approximately four amino acids per monomer.
The following figure supplement is available for figure 10:

**Figure supplement 1.** Phase separation of Nup98s from the remaining two species as well as of yeast nucleoporins ScNup100/ScNup116 from **Schmidt and Görlich (2015)** can also be accommodated within the theoretical model, with slightly different packing fraction and monomer volume.

## Ternary FG nup/transport proteins/buffer solutions

Phase separation in bulk solutions of FG nups and the transport proteins can be understood through the same theoretical prism. From the theoretical standpoint, at sufficiently high concentrations, such mixtures phase separate into a dilute phase in equilibrium with the dense phase, driven by the inter-chain cohesiveness and the interactions of the FG nups with the transport proteins. The fundamental physics behind this process is the same as in the pure FG nup phase separation: in the dense phase the magnitude of the energetic/enthalpic interactions is maximized at the expense of the entropy.

The overall predictions and conclusions of the theory are presented in the phase diagram in **Figure 12**. In a range of concentrations, the system phase separates into a dense phase of FG nups mixed with transport proteins, in equilibrium with a dilute solution. The corresponding phases lying on the boundary of the phase separation region (shown in black line) are connected with straight lines. The inset shows the effect of the transport protein-cargo complex size and their interaction strength with the FG nups on the phase separation. Increasing the interaction strength $\chi$ or decreasing the complex size $\bar{v}$ enhances the penetration of the transport proteins and into the dense phase and increases its overall density (green line). On the other hand, sufficiently big or too weakly interacting particles are excluded from the dense phase, as shown in the red phase separation line, where the density of the transport proteins is lower inside the dense FG nup phase than in the outside solution.

The model semi-quantitatively reproduces experimentally observed patterns of phase separation in bulk FG nup solutions mixed with different transport protein-cargo complexes as shown in

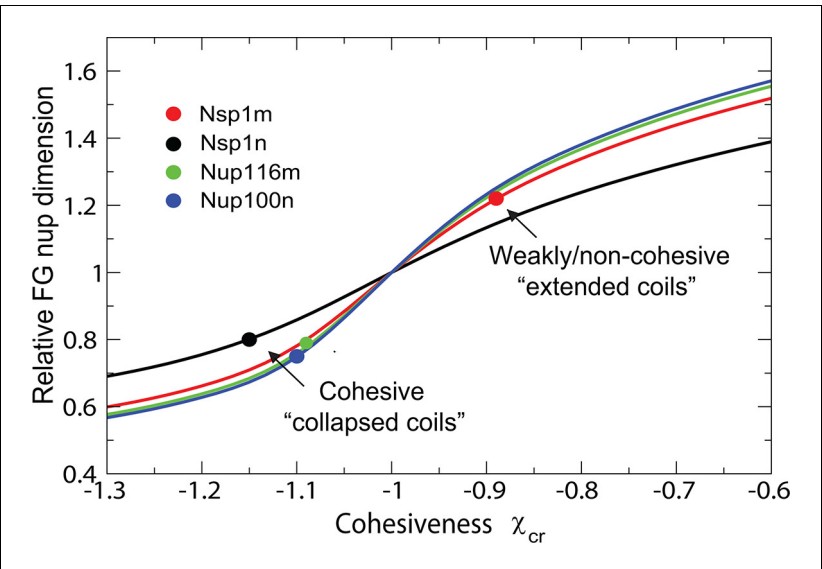

**Figure 11.** Dimensions of individual FG nucleoporins in solution: theory vs. experiment. Comparison of the model predictions for the dimensions of individual FG nups with the observations of Yamada et al (*Yamada et al., 2010*). Dimensions of the individual FG nup coils are normalized by the dimensions of the Gaussian chain of the same length ('relaxed coil' for the experimental data). Solid black, red, green and blue lines are the theoretical model predictions for $N = 43, 108, 137, 152$ corresponding to four amino acids per monomer for the respective FG nups. The model predictions are independent of the choice of the monomer size $v_0$. Circles: experimental measurements of the FG nup dimensions from *Yamada et al. (2010)*. Polymer model captures the bimodal distribution of 'extended' and 'collapsed' FG nups. This behavior, known as the coil-globule transition in polymer literature, reflects physically the same phenomenon as the decrease in the grafted layer height with increase in cohesiveness strength shown in *Figure 2*.

The following figure supplement is available for figure 11:

**Figure supplement 1.** Classification of individual FG nucleoporins according to their scaling exponent.

---

*Figure 13* for the mixtures of Nup98 with Importin-$\beta$-cargo complexes studied in *Schmidt and Görlich (2015)* (Importin-$\beta$ is the vertebrate homologue of the yeast Karyopherin-$\beta$1). Specifically, both NTF2 and Importin-$\beta$ penetrate the dense FG nup phase, but the complexes of Importin-$\beta$ with either medium (IBB-GFP) or large (IBB-MBP-GFP) cargo are excluded. By contrast, a very large complex of four Importin-$\beta$ molecules with four ZsGreen proteins does partition into the dense FG nup phase, showing that the large size exclusion is not absolute but can be overcome with stronger interactions. Other observations, such as the lack of penetration of the dense phase by the small TEV-mCherry (approx 25 kDa) complex, which does not interact with the FG nups ($\chi = 0$), are also captured by the model. The parameters for this analysis were based on those inferred from the analysis of phase separation of Nup98 in the absence of the transport proteins: N=167 $v_0 = 1$ nm$^3$, $\chi_{cr} = -1.5$. The approximate volumes of different constructs were estimated based on their molecular weights as explained in the Model section and *Figure 9*. The interaction parameters of Imp-$\beta$ with the FG nups are $\chi_{NTF2} = -75$, $\chi_{(Imp-\beta)} = -215$. To incorporate the potential direct repulsive interactions between the cargo and the FG nups described in *Schmidt and Görlich (2015)*, we assumed the interaction of the Imp-$\beta$-IBB-GFP and Imp-$\beta$-IBB-MBP-GFP complexes with the FG nups to be slightly weaker ($\chi = -200$). As a parsimonious estimate, we assumed that the interaction strength of the Imp-$\beta$-ZsGreen tetramer with the FG nups is four times that of an individual Importin-$\beta$. The correct penetration pattern is produced by the FG nup-Imp interaction strengths lying in the range $-220 < \chi_{(Imp-\beta)} < -200$, in a good agreement with the parameter values inferred from the analysis of the surface layers of Nup98 from Refs. *Wagner et al. (2015)* and *Kapinos et al. (2014)*; see *Figure 13—figure supplement 1*. Small differences in the values of $\chi$ between different

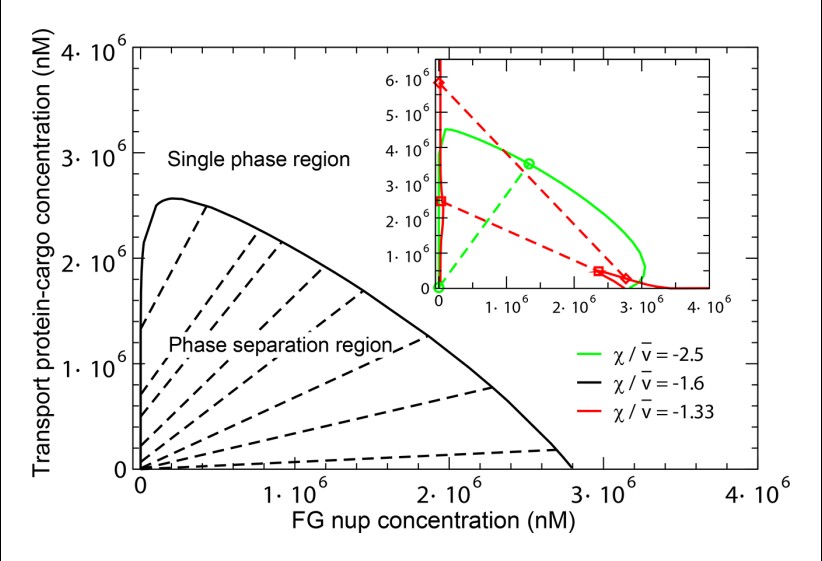

**Figure 12.** Phase separation in solutions of FG nups mixed with transport proteins: theoretical predictions. For high enough values of the ratio $\chi/\bar{v}$, mixtures of FG nucleoporins and the transport proteins phase separate into a dense aggregate containing both nups and transport proteins, and a dilute solution. The black line encloses the predicted phase separation region approximately corresponding to a mixture of Imp-$\beta$ and Nup98 calculated at $N = 167, \bar{v} = 125, \chi = -200, \chi_{cr} = -1.5$ and $\chi/\bar{v} = -1.6$. The oblique lines connect the coexisting phases that lie on the boundary of the region. Outside of this region, FG nups and transport proteins are homogeneously mixed and no phase separation occurs. *Inset*: Increasing the transport protein size $\bar{v}$ or decreasing $\chi$ hinders their penetration into the dense phase and eventually leads for their complete exclusion as shown in the red dashed lines connecting the co-existing phases: concentration of the transport proteins in the dense phase is lower than outside. Vice versa, increasing $|\chi/\bar{v}|$ enhances the penetration of the transport proteins into the dense phase as shown in the green lines.

The following figure supplement is available for figure 12:

**Figure supplement 1.** Phase separation region of *Figure 12* in a double logarithmic scale.

experimental systems are expected due to the differences in the buffers and the intrinsic sequence differences between the proteins fragments used in each study.

The model also predicts how the penetration of the transport proteins into the dense phase is affected by the cohesiveness $\chi_{cr}$, whose role in the NPC selectivity and permeability has been the subject of a vigorous debate (*Frey and Görlich, 2007*; *Schmidt and Görlich, 2015*; *Lim et al., 2007*; *2008*; *Wagner et al., 2015*; *Eisele et al., 2010*). At low cohesion, insufficient to cause the phase separation of the FG nups alone, addition of the transport proteins can trigger the phase separation and the formation of the mixed dense phase. Further increase in the cohesiveness *promotes* formation of an even denser phase and enhances Importin partitioning into it, contrary to the naive expectation that cohesiveness always inhibits particle penetration in to the FG Nup medium. Nevertheless, further increase in cohesiveness prevents penetration of the transport proteins into the dense phase, thereby enhancing the permeability barrier as has been suggested before (*Hülsmann et al., 2012*; *Frey and Görlich, 2007*). These results will be explored in detail elsewhere.

The model also might explain a recent report that the aggregation of cohesive FG nup segments can be reversed by addition of highly concentrated cell lysate (up to 50 mM) (*Hough et al., 2015*). Cell lysate has been independently shown to interact with the FG nups non-specifically but strongly enough to compete with the transport proteins for binding sites (*Tetenbaum-Novatt et al., 2012*). In our model this is captured through medium value of $\chi$, the interaction strength of the transport proteins with the FG nups. As can be seen from *Figure 12* and *Figure 12—figure supplement 1*, addition of a high concentration of attractive proteins to the phase separated FG nup aggregate can

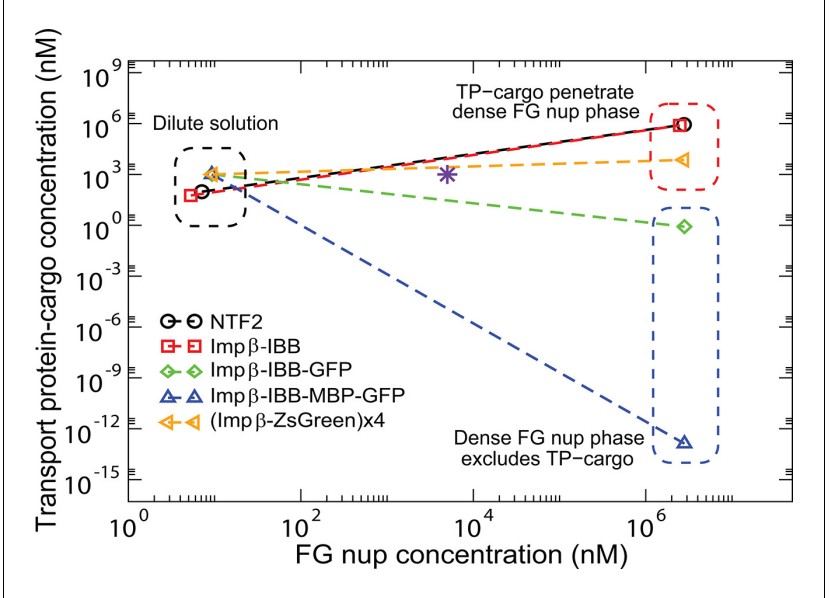

**Figure 13.** Transport protein partitioning into dense FG nup phase: theory vs. experiment. Theoretical model captures the pattern of partitioning of the transport proteins into the dense phase. Symbols connected with dotted lines show the theoretically predicted co-existing dilute and dense phases of TtNup98 at varying sizes of transport protein-cargo complexes corresponding to those used in *Schmidt and Görlich (2015)* and summarized in *Figure 9*. Transport proteins Importin-$\beta$ and NTF2 partition into the dense phase, while larger complexes of Importin-$\beta$ with different cargoes do not. Even a large cargo can be configured to penetrate the dense phase as illustrated by the example of a tetramer of Imp-$\beta$-ZsGreen complexes. All theoretical solutions are for the initial concentrations of 5 $\mu$M of Nup98A and 1 $\mu$M of the transport protein-cargo complexes (indicated by the pink star symbol). Parameters used to generate these solutions are established from the analysis of Nup98 phase separation in the absence of the transport proteins, described in *Figure 10*: N=167, $v_0$ = 1 nm$^3$, $\chi_{cr}$ = −1.5; see text for the interaction parameters of the transport proteins with the FG nups .

The following figure supplement is available for figure 13:

**Figure supplement 1.** Adsorption of Karyopherin-$\beta$1 into Nup98 layer and the layer height (inset) vs. bulk concentration of Kap-$\beta$1.

take the system out of the phase separation region, in agreement with the experimental observations.

## Discussion

Mechanistic understanding of transport through the Nuclear Pore Complex is hindered by the complexity of the NPC organization and the absence of experimental methods for directly probing FG nup conformations and dynamics during transport. One has to rely on the interpretation of indirect measurements of the FG nup properties in vitro, which are partially incomplete or conflicting. On the theoretical side, the progress is hampered by the lack of a universally accepted comprehensive theory of intrinsically disordered proteins that is able to incorporate all the factors dictating the physico-chemical and nanomechanical properties of the FG nups.

The theoretical model developed here provides a rigorous physical framework for organizing the known phenomenology of the observed FG nup behaviors and suggests ways to reconcile the apparently contradictory experimental findings and models of transport. The model relies on the minimal number of key physical concepts and variables to describe the FG nups and their interactions with the transport proteins, integrating the multitude of molecular details into a small number of variables, thus avoiding the over-fitting problem, inherent to models with multiple unknown parameters. It has been already suggested that the biophysical properties and dimensions of individual FG nups can be, to a large extent, captured in one phenomenological

parameter, the hydrophobic to charged amino acid content ratio, paralleling our cohesiveness strength $\chi_{cr}$ (*Yamada et al., 2010*). Our model extends these concepts into the regime of multiple chain assemblies interacting with transport proteins. We have found that this limited set of concepts is sufficient to qualitatively, and even semi-quantitatively, explain the experimentally observed patterns and trends. We expect that explicit inclusion of several important factors such as the spatial distributions of ions, discrete nature of the binding sites on the transport proteins, non-linear elasticity of the chains, potential heterogeneity of the FG nup sequence, and more careful modeling of the microscopic structure of the FG nups assemblies will lead to better quantitative agreement. Additional sources of discrepancy are the possible length polydispersity of the FG nups, transport protein aggregation, and inherent biases of the experimental techniques (for instance, in the SPR method, for technical reasons the measurement of the layer height lags behind in time after the measurement of the protein adsorption (*Wagner et al., 2015*). The theory provides clear experimental predictions for the variation of the properties of FG nup-transport protein assemblies with the experimental conditions such as the grafting density. Such measurements would allow further systematic refinement of the model.

The analysis of this paper shows that both entropic ('brush-like') and entalpic/cohesive ('gel-like') effects naturally cooperate in determining the spatial structures of the assemblies of FG nucleoporins with the transport proteins, and suggests how the 'brush' and the 'gel' concepts can be reconciled. Although the overall qualitative predicted behavior of the surface layers is similar for cohesive and non-cohesive chains alike, quantitative comparison with experiments was only possible by assuming a certain amount of cohesiveness for the studied FG nups. Our analysis indicates that all FG nups likely possess some degree of intra- and inter-chain cohesiveness, although only for some of them it is sufficiently strong to cause aggregation in bulk solutions. The model also shows that the different classes of 'extended' and 'collapsed' FG nups (*Yamada et al., 2010*) can be accounted for by different values of the cohesiveness $\chi_{cr}$. Another puzzling observation of the resilience of the NPC transport with respect to the deletion of large numbers of FG nups might be attributed to the fact that the permeability and the selectivity of the FG nup assemblies are relatively insensitive to the grafting density in a significant range - at lower grafting densities, the neighboring chains simply expand, maintaining selective permeability properties of the layer (*Popken et al., 2015*). Finally, relatively weak interaction energies of the FG nups among themselves and with the transport proteins inferred from our analysis are consistent with the high local molecular mobility inside the dense aggregates observed experimentally in *Schmidt and Görlich (2015)*, *Hough et al. (2015)*.

The success of the theory relies on the very robust physical mechanisms underlying it. Attractive interactions between long flexible filaments and compact objects, such as folded proteins, cause their surface assemblies to attain more compact conformations at low concentrations of the transport proteins and swollen conformations at higher concentrations. Similarly, in bulk solutions of ungrafted chains, inter-chain interactions and the interactions with transport proteins lead to phase separation and formation of a dense phase at sufficiently high concentrations. These behavior motifs are always expected irrespective of the nature of the interactions on the molecular scale. In this sense, all such systems lie in the same 'universality class' (*de Gennes, 1979*). The ability of the model to capture the behavior of different FG nups in different geometries under a variety of experimental conditions makes it a useful tool for the development of further, more refined, models. Finally, the analysis of this paper underscores the importance of always considering the presence of the transport factors when thinking about NPC architecture.

The theory also sheds light on the discrepancies in the experimental measurements of the binding affinities of the FG nups to transport proteins. Depending on the experimental procedure, the values of the measured dissociation constants range from several nanomolars to several micromolars. Moreover, some of these measured affinities appear to be inconsistent with the observed transport times in the millisecond range (*Yang and Musser, 2006*; *Yang et al., 2004*; *Tetenbaum-Novatt and Rout, 2010*; *Tetenbaum-Novatt et al., 2012*; *Tu et al., 2013*; *Denning et al., 2003*; *Ma, 2010*). Our theory shows how these discrepancies might stem from the fundamental statistical thermodynamics of the transport protein-FG nup interaction. First, as shown in this paper, penetration of the transport proteins into an FG nup layer is a cooperative process, not described well by a single Langmiur isotherm typically used in the interpretation of binding assays. Second, penetration of the transport proteins into the layer is determined not only by enthalpic but also entropic effects and therefore the measured effective affinity does not directly reflect the interaction energies. Finally,

even the purely enthalpic part of the interaction can vary with the experimental conditions, because the average number of monomers available to bind to a transport protein depends on the monomer concentration, which in turn depends on the layer height and grafting distance (*Kapinos et al., 2014*; *Schoch et al., 2012*; *Tu et al., 2013*; *Sethi et al., 2011*). All this highlights the fact that the classical characterization of inter-molecular interactions by a single affinity value is not informative for complex multivalent interactions of spatially extended objects such as the FG nups. It is also worth bearing in mind that a $1000$-fold difference in the dissociation constant translates into $\approx 7kT$ difference in the effective interaction strength - a relatively small difference that can be easily influenced by many factors.

The results of this paper have important implications for the behavior of the Nuclear Pore Complex and design of bio-molecular sorters based on the same principles. One has to be careful in making inference about the NPC properties based on the in vitro results because the detailed features of the actual spatial morphologies of FG nup assemblies in the channel-like geometry of the NPC are likely to differ from flat and bulk geometries. Analysis of this paper establishes the pertinent parameters that constrain possible scenarios and guide future model building. Recent work on coarse grained models of flexible chains in channel geometries allows us to gauge the implications of the findings in the flat and bulk geometries for FG nup morphologies within the NPC (*Osmanović et al., 2013a*; *Coalson et al., 2015*; *Peleg et al., 2011*). Behavior of non-cohesive or weakly cohesive chains in relatively wide pores - wider than the natural height of the planar layer - is expected to be qualitatively similar to that of flat layers. In this case, the chains form a relatively dense layer along the inner surface of the channel, and increase in cohesiveness compacts the layer towards the walls. Addition of transport proteins causes either collapse or swelling of this surface layer, analogously to the behavior in the flat geometry. In the other limit of strongly cohesive chains in relatively narrow channels, the monomers of the chains accumulate at the pore center in a plug-like shape. It is still unknown in what regime the NPC lies, and further computational and experimental work is required. These findings are also interesting in the more general context of intrinsically disordered proteins, where physics concepts are often invoked to organize and explain the experimental observations (*Uversky, 2002*; *van der Lee et al., 2014*; *Das et al., 2015*).

Although the qualitative behavior motifs predicted by the theory and borne out by the experiments are very general, the specific quantitative features such as the exact height or the degree of compaction are rather sensitive to the parameter values, such as the grafting distance, the density and the interaction strength. This has been already noted in other computational theories (*Popken et al., 2015*; *Osmanović et al., 2013a*; *Tagliazucchi et al., 2013*; *Gamini et al., 2014*). This raises a question - how the Nuclear Pore Complex functioning remains so invariant across species despite the large variations in sizes and spatial organization. Similarly puzzling in this light is the ability of the NPC to maintain its function despite large structural perturbations (*Strawn et al., 2004*; *Hülsmann et al., 2012*). One possible solution to this puzzle is that the NPC is exquisitely "fine tuned" in a sense that it works correctly only when all the molecular details are right - the FG nup sequence and localization, local pH, ionic strength and concentrations of other molecules - and these conditions are maintained by the cellular homeostasis. On the other hand, the results of this model suggest another possibility - that the NPC is 'robust' in a sense that any structure with the approximately right physical properties will function nearly optimally, which could explain the NPC resilience with respect to structural damage and re-arrangements. This provides an interesting example of functional conservation in the absence of sequence conservation. The importance of this question goes beyond the NPC and arises in the discussion of many cellular machines and networks (*Alon et al., 1999*; *Bialek, 2012*).

## Acknowledgements

AZ acknowledges the support of the National Science and Engineering Research Council of Canada (NSERC) and Compute Canada. RDC acknowledges support form NSF grant CHE-1464551. RYHL is supported by the Swiss National Science foundation through Grant 31003A-146614. We thank Dr. B Hoogenboom, Dr. S Musser, Dr. P Onck, Dr. R Richter and Dr. R Schoch for helpful discussions.

## Additional information

### Funding

| Funder | Grant reference number | Author |
| --- | --- | --- |
| National Science and Engineering Research Council of Canada | Discovery Grant RGPIN 4025901 | Andrei Vovk<br>Anton Zilman<br>Chad Gu |
| Swiss National Science Foundation | Grant 31003A_146614 | Larisa E Kapinos<br>Roderick YH Lim |
| National Science Foundation | CHE-1464551 | Rob D Coalson |

The funders had no role in study design, data collection and interpretation, or the decision to submit the work for publication.

### Author contributions

AV, Performed the calculations, Analysis and interpretation of data, Drafting or revising the article; CG, RDC, DJ, Conception and design, Analysis and interpretation of data, Drafting or revising the article; MGO, Performed the calculations, Conception and design; LEK, RYHL, Acquisition of data, Analysis and interpretation of data, Drafting or revising the article; AZ, Performed the calculations, Conception and design, Analysis and interpretation of data, Drafting or revising the article

### Author ORCIDs

Anton Zilman, http://orcid.org/0000-0002-8523-6703

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

## Appendix

### Details of the phase separation calculations

#### Two-component (FG nup/buffer) solutions

In a system where the only solute are the FG Nups without the transport proteins, phase separation occurs for sufficiently high $|\chi_{cr}|$. The volume fractions of the monomers $\psi_1$ and $\psi_2$ in the co-existing phases are determined from the equality of the chemical potentials $\mu$ and the osmotic pressures $\Pi$ in the separated phases:

$$
\begin{aligned}
0 &= \mu_\psi(\psi_1) - \mu_\psi(\psi_2) = \frac{df}{d\psi}\Big|_{\psi=\psi_1} - \frac{\partial f}{\partial \psi}\Big|_{\psi=\psi_2} \\
0 &= \Pi(\psi_1) - \Pi(\psi_2) \\
\Pi(\psi) &= -f(\psi) + \psi\frac{df}{d\psi},
\end{aligned}
\tag{1}
$$

where the free energy $f(\psi)$ is given by **Equation (4)** in the main text with $\phi = 0$. Numerical solution of these equations produces the phase diagram of **Figure 10** in the main text.

#### Three-component (FG nups/transport proteins/buffer) solutions

In mixed solutions of the FG nups and transport proteins, the chemical potentials of each species in the coexisting phases are equal to each other, as is the osmotic pressure in both phases, similar to the two component case above (**de Gennes, 1979**):

$$
\begin{aligned}
0 &= \frac{\partial f}{\partial \psi}\Big|_{\psi,\phi=\psi_1,\phi_1} - \frac{\partial f}{\partial \psi}\Big|_{\psi,\phi=\psi_2,\phi_2} \\
0 &= \frac{\partial f}{\partial \phi}\Big|_{\psi,\phi=\psi_1,\phi_1} - \frac{\partial f}{\partial \phi}\Big|_{\psi,\phi=\psi_2,\phi_2} \\
0 &= \Pi(\psi_1,\phi_1) - \Pi(\psi_2,\phi_2) \\
\Pi(\psi,\phi) &= -f(\psi,\phi) + \psi\frac{\partial f}{\partial \psi} + \phi\frac{\partial f}{\partial \phi}.
\end{aligned}
\tag{2}
$$

However, unlike in the two component solution, three **Equation (2)** in themselves are not sufficient to determine the four unknowns $(\psi_1,\phi_1)$ and $(\psi_2,\phi_2)$. In the ternary solutions FG nups/transport proteins/buffer, the densities of both components in the coexisting phases depend also on the initial concentrations $\phi$ and $\psi$ that satisfy the following conservation laws:

$$
\begin{aligned}
\psi_1(1-r) + \psi_2 r &= \psi \\
\phi_1(1-r) + \phi_2 r &= \phi,
\end{aligned}
\tag{3}
$$

where $r = (V - V_2)/V$ is the fraction of volume occupied by the phase with $(\psi_2,\phi_2)$.

Together, **Equation (2-3)** provide five equations for five variables $(\psi_1,\phi_1)$, $(\psi_2,\phi_2)$ and $r$. Their solution results in the phase diagrams of **Figures 11–12** in the main text.

