## [Decision Letter]

Thank you for submitting your work entitled "Simple biophysical model explains the conformational transitions of the unfolded proteins of the Nuclear Pore Complex" for consideration by *eLife*. Your article has been reviewed by three peer reviewers, and the evaluation has been overseen by a Reviewing Editor and John Kuriyan as the Senior Editor.

The reviewers have discussed the reviews with one another and the Reviewing editor has drafted this decision to help you prepare a revised submission.

This paper develops a self-consistent-field theory for a model of transport through nuclear pores. The model is properly formulated based on well-established and long-standing concepts in polymer physics. Though several theoretical models have been previously published attempting to describe the biophysical nature of nuclear transport, the model presented here has the considerable attraction that it is less inclined than these other models to try to pitch a particular mechanism – rather, it sets up a computational framework that can be adapted and added to upon additional data, giving researchers a computational framework for a more rigorous analysis of data. To demonstrate the effectiveness of the model, it is used to perform a new set of analyses of previously published work from two of the co-authors, Kapinos and Lim. The model re-capitulates certain previously puzzling behaviors seen for FG nups and provides a biophysical hypothesis for these behaviors. This model could therefore be a valuable new tool. However, our enthusiasm for the paper is considerably diluted by two issues.

Firstly, the manuscript oversells itself as having resolved the controversies, and furthermore, the comparisons with experiment are qualitative. Qualitative comparisons are fine, but rephrasing some of the text so that it accurately reflects the accomplishments is essential.

Secondly, and more importantly, to establish itself as a useful computational platform, additional work is necessary. In the paper, the model is fit to existing data from some of the co-authors. To make a compelling case that the model indeed captures the essential features of nuclear pore transport, it is necessary to show that the model is predictive – that is, take published data from several groups (not the co-authors') that was explicitly not included in building the model, and without re-adjusting the parameters used to fit the authors' data, show that the model describes these additional data. Without demonstrating this capability, the model cannot be considered predictive or explanatory. We suggest that you consider describing the data presented in the following references: 1) Strawn et al., 2004: can the model describe the observation here that so much of the FG mass can be deleted without completely disrupting transport? 2) Yamada et al.: can the model explain why might there are two classes of cohesiveness as suggested in this paper? 3) Popken et al., 2015: Compare the coarse grained simulations and data in this paper with those described in the manuscript. 4) Patel et al., 2007: which also contains interesting data. Of course, you could choose other data sets that you believe are illustrative.

Some additional detailed points that need to be addressed are listed below.

1) The model completely ignores sequence information and electrostatic interactions. The problem with this is that there is clear evidence from bioinformatics studies (see ref. 68) and theory (see ref. 50) that clearly show the need to consider the role of electrostatics. These studies seem to consider similar screening conditions as that in the present manuscript. Moreover, ref. 50 (a theoretical study) shows that smearing the sequence leads to qualitatively different behavior. Therefore, these points need to be mentioned explicitly as refs. 68 and 50 suggest that it may be difficult to describe the NPC using minimal models adequately, as is claimed in the last sentence of the manuscript.

2) If the minimal model provides an excellent description of nuclear pores, why can synthetic systems not recapitulate the behavior of nuclear pores? The manuscript should emphasize that the model describes some universal features, and more details will have to be added upon considering additional data – this point might become more vivid or refuted by addressing the second major point made above.

3) Figure 2 presents well known results, as is noted in the manuscript, but the original work of Alexander is not cited. Also use of the term "cross-link" here is misleading since this is just a model van der Waals interaction.

4) Figure 3 shows the two limits that a brush can have from maximal repulsion (stretching) to completely compact. The fact that the experimental observations fall between these two asymptotic limits is expected. But, why is the same nup at different grafting densities suggesting very different values of *χ*?

5) It appears that the model assumes macroscopic uniform planar layers grafted with one type of FG repeat. Does the model also make accurate predictions for geometries that more accurately resemble the NPC i.e. nanoscopic pores, or mixtures of different FG repeat types?

[Editors' note: further revisions were requested prior to acceptance, as described below.]

Thank you for resubmitting your work entitled "Simple biophysics underpins collective conformations of intrinsically disordered proteins of the Nuclear Pore Complex" for further consideration at *eLife*. Your revised article has been favorably evaluated by John Kuriyan as Senior editor, a Reviewing editor, and one reviewer. The revised manuscript is much improved, and most of the points we raised previously have been fully addressed. If the remaining points detailed below can be addressed, we think that there is a very good chance that the paper will be accepted for publication.

1) In the Abstract, we still feel "These results reconcile some of the outstanding controversies…" is strong, given that this work does not concretely reconcile, but rather suggest solutions, to the current controversies. Perhaps "These results address some of the outstanding controversies…" would be better.

2) In the Discussion, the sentence "The suggests how the ‘brush’ and… effects" is grammatically incorrect and unclear – can you please re-write?

3) On use of data by others: You addressed our comments by including a work that focuses only on Nup98, which represents only one type of FG, one that can form gels in vitro, but are not necessarily representative of others. Even if an experiment is not possible, could you still replicate computational findings by others, if not at a quantitative, at a qualitative level? In the absence of an independent test, again, we would recommend a more cautious approach on claims that the current work resolves different findings. Instead, perhaps emphasize rather that this work presents a tool of utility to future studies.

4) On the omitting of some aspects of FGs, etc.: A) Sequence: the explanation about the randomized AA sequence is much improved. However, it would be nice to have a demonstration of the thesis in a case with a simple heterogeneity in sequence, such as in Nsp1 (i.e. difference in property in N-terminal and C-terminal regions). Such a demonstration would also reinforce the paper by possibly recapitulating the computational finding in Yamada et al. (difference in expected structure on N and C-termini). B) On the geometry of FG grafts: Although further computational investigation of this topic may be beyond the scope of this paper, it is worth discussing the limitations of the current presentation of the simple monolayer geometry and the possible differences/changes in FG behavior/conformation (and/or lack thereof) that could arise from change in geometry (e.g. grafting them in a pore lumen, etc.).

5) In the Abstract, there is a sentence: "NPC transport relies on the conformational transitions of assembly…". It is true that the conformation is known to change in in vitro experimental setups, but is there any evidence in NPC-context that supports this statement (i.e. conformational transitions)?

6) Figure 1 is rather crudely rendered, and does not accurately represent the positions of the FG nups indicated. Krull et al., 2004 can be used for Nups98 and 153, while recent crystallographic work (Chug et al., 2015) indicates the position of Nup62 to be symmetric and immediately adjacent to Nup93 (Krull et al., 2004). Also, the lower two panels as they stand don't really help readers understand what you call the "experimental situation." This figure should be polished further.

7) For Figure 2, it can be guessed from the context that different curves correspond to different *χ*s, but it's better to specify which curve correspond to what value (or add some direction like an arrow with "increasing *χ*," etc.)

8) For Figure 3, please consider omitting "experimental evidence of cohesiveness" from the figure title. Strictly speaking, it should go something like "polymer model based on cohesion theory fits well with the data" and it's not really evidence for cohesiveness.

---

## [Author Response]

*[…] Firstly, the manuscript oversells itself as having resolved the controversies, and furthermore, the comparisons with experiment are qualitative. Qualitative comparisons are fine, but rephrasing some of the text so that it accurately reflects the accomplishments is essential.*

We have modified the text in the Introduction, the Discussion, as well as the title and the Abstract accordingly to reflect more accurately the results and the achievements of the paper and their relation to the outstanding experimental puzzles. In particular, we feel that the model provides an important insight into the “gel” vs. “brush” controversy by demonstrating that the entropic and the cohesive factors are not mutually exclusive but naturally cooperate in determining the properties of FG nup assemblies.

*Secondly, and more importantly, to establish itself as a useful computational platform, additional work is necessary. In the paper, the model is fit to existing data from some of the co-authors. To make a compelling case that the model indeed captures the essential features of nuclear pore transport, it is necessary to show that the model is predictive* – *that is, take published data from several groups (not the co-authors') that was explicitly not included in building the model, and without re-adjusting the parameters used to fit the authors' data, show that the model describes these additional data. Without demonstrating this capability, the model cannot be considered predictive or explanatory. We suggest that you consider describing the data presented in the following references: 1) Strawn et al., 2004: can the model describe the observation here that so much of the FG mass can be deleted without completely disrupting transport? 2) Yamada et al.: can the model explain why might there are two classes of cohesiveness as suggested in this paper? 3) Popken et al., 2015: Compare the coarse grained simulations and data in this paper with those described in the manuscript. 4) Patel et al., 2007: which also contains interesting data. Of course, you could choose other data sets that you believe are illustrative.*

We thank the editors and the reviewers for this very important comment. In order to address it, we chose to compare our model with the experimental data from the following paper: Schmidt and Görlich, "Nup98 FG domains from diverse species spontaneously phase-separate into particles with nuclear pore-like permselectivity." *eLife* 4 (2015): e04251. We chose this work because it contains extensive and systematic study of the morphologies of in vitro assemblies of FG nucleoporins with nuclear transport proteins. Unlike the previous “gels”, in this work, the FG nup aggregates with the transport factors are formed through an equilibrium phase separation mechanism, allowing meaningful comparison with our model; this study parallels the experimental work on FG nup monolayers. The chosen paper provides semi-quantitative data that can be meaningfully and informatively compared with the theoretical predictions in order to validate or refute the model. It also satisfies the other requirements of the reviewers: the experiments are done in a different geometry – bulk solutions rather than in surface layers – and by a different group, using entirely different experimental techniques (see below for additional reasons for choosing this work as opposed to other experimental work).

The chosen work investigates the phase separation of FG nups (specifically, Nup98 and its yeast homologues) and the penetration of Importins into the dense FG nup phase. The theoretical model predictions are in excellent agreement with the experimental data for parameter values that are very close to those describing the behavior of Nup98 with Karyopherins in grafted surface layers. However, we emphasize that the validity of the model is not determined by parameter fitting, but rather through addressing clear cut trends and patterns – such as the dependence of the Importin penetration into the dense phase on the cargo size. Small differences in parameter values (as long as they lie in the relevant physical regime) are expected and can be attributed to a number of factors – inherent molecular differences between Kap-*β* and Imp-*β*, different buffers (as a case in point, the buffers in the work of Lim and coworkers do not contain denaturant, while those used by Schmidt and Görlich do) and differences between the FG nups from different species.

We have expanded the Model section to include the description of bulk solutions of FG nups mixed with transport factors and have added a whole new section in the Results, titled “Phase separation in bulk solutions of FG nups mixed with transport proteins”, which describes the comparison of the theoretical model with the data, including several new figures, Figure 9, Figure 10, Figure 11 and Figure 12 and corresponding Figure Supplements that also include the sensitivity analysis with respect to the parameter choices. We have also modified the Introduction and the Discussion sections accordingly. We have also expanded the Supplementary Text with the details of the theoretical calculations of the phase separation.

We have added text in the Discussion regarding the relation of our model to other experimental works suggested by the editors and the reviewers. Unfortunately, the suggested works are less suitable for meaningful theoretical analysis. The data in Patel el al. Cell (2007) are qualitative; Yamada et al. Mol. Cell. Prot. (2010) does not provide quantitative information about the interactions of Karyopherins and FG nups while FG nup dimensions are determined through size fractionation, analysis of which is beyond the scope of the present work. The experimental work by Strawn et al. Nat. Cell. Biol. (2004) and the subsequent work by Popken et al. Mol. Biol. Cell (2015) deal with a much more complex situation in vivo that could be affected by a number of factors that are not included in our model. The computational model used in Popken et al. does not consider the effect of the Karyopherins on the FG nup conformations and has been parameterized for yeast FG nucleoporins, precluding quantitative comparison. Nevertheless, our model is consistent with the findings of all these works. For instance, the two classes of the cohesive and non-cohesive FG domains reported in Yamada et al. can be accommodated through different values of the cohesiveness strength *χ_cr_*. Robustness of the NPC transport with respect to FG nup deletions observed in Strawn et al. and Popken et al. can potentially be attributed to the fact that the flexible FG nup chains entropically swell at lower densities (as already been suggested by us in Zilman et al., PLoS Comp Biol 2007). Also, we have also recently reported results of our coarse-grained model in the NPC-like geometry in Coalson et al., JPC(2015) predicting polymer density distributions similar to those described in more detailed models of Onck and collaborators and Szleifer and collaborators. Thus, our model agrees with the morel detailed models (at least qualitatively) in the regime of its validity. However, we feel that the full discussion of the detailed model predictions for the NPC transport is beyond the scope of the present work, and will be performed in future works.

*Some additional detailed points that need to be addressed are listed below. 1) The model completely ignores sequence information and electrostatic interactions. The problem with this is that there is clear evidence from bioinformatics studies (see ref. 68) and theory (see ref. 50) that clearly show the need to consider the role of electrostatics. These studies seem to consider similar screening conditions as that in the present manuscript. Moreover, ref. 50 (a theoretical study) shows that smearing the sequence leads to qualitatively different behavior. Therefore, these points need to be mentioned explicitly as refs. 68 and 50 suggest that it may be difficult to describe the NPC using minimal models adequately, as is claimed in the last sentence of the manuscript.*

We thank the referees for these crucial points and we regret that they were not clearly explained in the previous version of the paper.

Electrostatics: Although we do not explicitly model the ion distributions as in, for instance, Tagliazzucci et al., PNAS (2013), the electrostatic interactions are included in the model in the same fashion as in other computational models of the FG nups (Ghavami et al., BJ 2014) and other unfolded proteins (Leermakers, Soft Matter, 2010) – through the screened electrostatic interaction. Electrostatics contributes to the interaction parameters *χ* and *χ_cr_*, along with all other interactions, such as the hydrophobic, π-π and others. This is a commonly used and well established approximation, which has been successfully applied to a variety of biopolymers. For the system under study, modeling the screened electrostatic interaction as a short range potential is a reasonable approximation because the Debye screening length is about 1 nm in the experimentally used salt concentrations, comparable to the size of the chain “monomers”. We note in passing that the microscopically realistic modeling of electrostatics of proteins in ionic solutions is extremely challenging because, among other things, soluble ions and charged side chains affect the hydrogen bonding structure of water and hence affect the hydrophobic interactions; at present all existing models employ various approximations, to the best of our knowledge. We added text in the Model section and the Discussion elucidating this point.

Sequence: Crucially, our model does not correspond to a randomized FG nup sequence, such as those discussed in the computational models of Tagliazucchi et al. PNAS (2013) and Ghavami et al., BJ (2014), and does not contradict their findings indicating that the sequence could significantly affect the behavior of FG nup assemblies. In all comparisons with the experimental data, we use a “monomer” of approximately four amino acids, which corresponds to an XXFG patch on an FG nup. The sequence motif of Glycin (G) following Phenylalanine (F) is known to be structurally important for the FG nup binding to the binding pockets on the transport proteins (for instance, Bayliss et al. JBC 2002 and Bayliss et al. Cell 2000). Randomization of the FG nup sequence would destroy the binding of FG nups to Karyopherins/Importins or make it much weaker, which in our model would correspond to lower interaction parameters *χ* and *χ_cr_*. We have modified the text in the Model to make this point clearer and added several new references.

*2) If the minimal model provides an excellent description of nuclear pores, why can synthetic systems not recapitulate the behavior of nuclear pores? The manuscript should emphasize that the model describes some universal features, and more details will have to be added upon considering additional data* – *this point might become more vivid or refuted by addressing the second major point made above.*

We are in complete agreement with the referees. Synthetic pores – whether using the actual FG nups or synthetic polymers – do not fully reproduce the behavior of the NPC. However, they do recapitulate many aspects of its selectivity – in particular discrimination between the transport proteins and the neutral molecules (Jovanovic-Talisman etal, Nature 2009, Kowalczyk, Nat. Nanotech. 2011, Caspi and Elbaum, Nano Lett 2008 and others). Indeed, these results were one of the motivations for our work because they suggest certain universal features that can be encapsulated in a relatively coarse-grained model. Further details of the NPC transport – such as the multiple FG nup types and RanGDP, to mention a few – need to be introduced into the model to capture the full complexity of NPC architecture and transport. In this respect, our model provides constraints and guides the inclusion of the important molecular details in more detailed theories. We revised the text in the Introduction and Conclusions to emphasize this point and added several new references.

*3) Figure 2 presents well known results, as is noted in the manuscript, but the original work of Alexander is not cited. Also use of the term "cross-link" here is misleading since this is just a model van der Walls interaction.*

We apologize for this oversight. We have added the reference to the original Alexander paper, as well as to the later work by Sanchez and Haran. We removed the term “cross-link” altogether, and stick to “cohesiveness” throughout the paper. We have rewritten the text in the Model section to explain that *χ_cr_*includes all interactions between the chains, including van der Waals, hydrophobic, electrostatic and other possible interactions such as π-π or π-charge interactions.

*4) Figure 3 shows the two limits that a brush can have from maximal repulsion (stretching) to completely compact. The fact that the experimental observations fall between these two asymptotic limits is expected. But, why is the same Nup at different grafting densities suggesting very different values of* χ*?*

We are in complete agreement with the referee that this behavior is expected on the polymer physics grounds. However, it was not a priori clear that the grafted FG nup layer can be well described by the polymer brush theory. A number of factors could have led to deviations from this polymer brush behavior – for instance, significant residual secondary structure. This figure indicates that the polymer concepts are applicable to this complex system and serves as the stepping stone for further model with Karyopherins. We revised the text to enhance this point.

If we understand correctly the second part of the question – the value of *χ_cr_* is not determined from the brush height at one value of the grafting distance but from the exponent *g* that characterizes the dependence of the layer height *h* on the grafting distance *a*, h~a−g.The values of *χ_cr_* and *χ* are indeed different for different FG nups, reflecting the different sequence and the hydrophobic vs. charge content. We have re-plotted Figure 3 and have rewritten the corresponding text to clarify this point.

*5) It appears that the model assumes macroscopic uniform planar layers grafted with one type of FG repeat. Does the model also make accurate predictions for geometries that more accurately resemble the NPC i.e. nanoscopic pores, or mixtures of different FG repeat types?*

We thank the referee for raising this point. At the moment there are no quantitative data for mixed layers or quantitative experiments in vitro in an NPC-like geometry that could be directly compared with the model. Nevertheless, simulations of the coarse grained representation of the FG nups in the NPC-like geometry (Coalson et al. JPC, 2015) show that the model predictions for the FG nup distribution in such geometry are in general agreement with the more molecular coarse-grained models of Taglizaucchi et al. PNAS 2013 and Ghavami et al. BJ 2014. However we feel that further discussion beyond the scope of the present work, and we intend to investigate it in detail in future work.

[Editors' note: further revisions were requested prior to acceptance, as described below.]

*1) In the Abstract, we still feel "These results reconcile some of the outstanding controversies*…*" is strong, given that this work does not concretely reconcile, but rather suggest solutions, to the current controversies. Perhaps "These results address some of the outstanding controversies*…*" would be better.*

We revised the Abstract accordingly.

*2) In the Discussion, the sentence "The suggests how the ‘brush’ and*… *effects" is grammatically incorrect and unclear* – *can you please re-write?*

We have rewritten that part of the Discussion.

*3) On use of data by others: You addressed our comments by including a work that focuses only on Nup98, which represents only one type of FG, one that can form gels in vitro, but are not necessarily representative of others. Even if an experiment is not possible, could you still replicate computational findings by others, if not at a quantitative, at a qualitative level? In the absence of an independent test, again, we would recommend a more cautious approach on claims that the current work resolves different findings. Instead, perhaps emphasize rather that this work presents a tool of utility to future studies.*

*4) On the omitting of some aspects of FGs, etc: A) Sequence: the explanation about the randomized AA sequence is much improved. However, it would be nice to have a demonstration of the thesis in a case with a simple heterogeneity in sequence, such as in Nsp1 (i.e. difference in property in N-terminal and C-terminal regions). Such a demonstration would also reinforce the paper by possibly recapitulating the computational finding in Yamada et al. (difference in expected structure on N and C-termini).*

We thank the referee and the editors for suggesting this. We have revised the text along the suggested lines to more accurately frame our results in the context of previous works. In addition, we have analysed the data from Yamada (2010) using our model and find a very good agreement.

We regret that the assumptions of our model and its relation to previous works were not explained clearly. In fact, our treatment of FG nup cohesiveness is conceptually similar to the analysis of Yamada et al. (2010). That work showed that a single coarse-grained phenomenological parameter could capture the cohesiveness of diverse FG nups. Specifically, FG nup dimensions and biophysical properties were categorized into two distinct classes according to the ratio of the number of hydrophobic to the number of charged amino acids in the FG nup sequence. Our model puts this notion onto a rigorous physical footing, quantitatively defining the cohesiveness via by the parameter *χ_cr_*. We stress that this treatment does not assume a randomized sequence but rather integrates the effects of the amino acid composition on the cohesiveness into one model parameter, *χ_cr_*. The main focus of our work is the extension of this approach to systematic analysis of the spatial structures of the assemblies of multiple FG nups and their interactions with the transport proteins.

To put our model in context and to further examine its predictive abilities, we analysed the data in Yamada (2010) from the vantage point of our model, focusing on Nsp1m, Nsp1n, Nup116m and Nup110n. We chose these segments because they are representative of the two classes of FG nups reported in Yamada (2010): the “cohesive” collapsed coils – such as the N terminal domain of Nsp1, (Nsp1n), Nup116m, Nup110n – and the “non-cohesive” extended coils, such as the C terminal domain of Nsp1, Nsp1m. Moreover, this choice enables direct comparisons with grafted planar layers of Nsp1 used in Wagner et al. (2015), which share an overlapping sequence with Nsp1m, grafted layers of Nup98 used in Kapinos et al. (2014) and Nup98 in bulk solutions used by Schmidt and Görlich (2015). Nup116 and Nup100 are yeast analogues of Nup98 and share similar biophysical and functional properties.

Our results are summarized in the new Figure 11 and its Supplement. Briefly, our analysis recapitulates the bimodal distribution of Yamada et al. by using different values of *χ_cr_* to characterise the collapsed coils (Nsp1n, Nup100n, and Nup116m) and the “extended coils” (Nsp1m). The values of *χ_cr_* arising from the analysis of the Yamada et al. data are entirely consistent with those obtained from the analysis of grafted planar layers of Nsp1 in Wagner et al. (2015), Kapinos et al. (2014) and of Nup98 in Kapinos (2014) and the cohesive bulk solution of Nup98, Nup110, Nup116 in Schmidt and Görlich (2015) barring small quantitative differences in *χ_cr_* that can be attributed to intrinsic sequence variations between the used FG nup segments, different buffer conditions and inherent difficulties in inferring chain dimensions from hydrodynamic radii in Yamada et al.

Consequently, our model is in agreement with other computational models in the regimes where they agree with each other and were compared with the experimental data. For instance, all theories agree that the Nup98/Nup100/Nup116 is more “cohesive” than the C terminal domain of Nsp1. However, current models in the literature lack consistency in their parameterizations as well as in their predictions. For instance, Taglizaucci et al. (2013) predict dimensions of the FG nup coils that are much larger than those predicted by Onck and coworkers, Gamini and Schulten (2014) report formation of FG nup “braids” while the model of Ghavami and Popken does not show those, etc. In light of these discrepancies, our goal was the development of a quantitative model that potentially can explain and reconcile FG Nup behavior observed in independent experiments and computational studies.

To summarize, with these new results, our theory accounts for the experimentally observed behavior of both cohesive and non-cohesive FG nups and their assemblies (alone and with transport proteins of various sizes). Moreover, model parameterization is consistent across different geometries (planar layers and bulk solutions) and buffers. To the best of our knowledge, this is the most extensive comparison of a theoretical model with experimental data. Hence, we believe that it forms an appropriate basis and a useful tool for future studies of the spatial architectures of the FG nups in various geometries. We modified the text in line with the referee and the editor requests to reflect these points more accurately.

*B) On the geometry of FG grafts: Although further computational investigation of this topic may be beyond the scope of this paper, it is worth discussing the limitations of the current presentation of the simple monolayer geometry and the possible differences/changes in FG behavior/conformation (and/or lack thereof) that could arise from change in geometry (e.g. grafting them in a pore lumen, etc).*

We thank the referee and the editors for this comment. We have added the appropriate commentary in the Discussion.

*5) In the Abstract, there is a sentence: "NPC transport relies on the conformational transitions of assembly…". It is true that the conformation is known to change in in vitro experimental setups, but is there any evidence in NPC-context that supports this statement (i.e. conformational transitions)?*

This is a good point. Indeed, direct exploration of the conformational changes of the FG nup assemblies within the NPC in vivo is difficult, due to the lack of appropriate experimental methods. However, most of the models (with the possible exception of the virtual gating theory) postulate such transitions based on in vitro observations, generating considerable controversies. Our goal was to address some of these controversies and clarify the salient physical factors governing the behavior of FG nup assemblies with transport proteins.

Nevertheless, there is some direct evidence of conformational changes of the FG nup assemblies in the NPC: Ma et al., PNAS, (2012) showed that addition of Kap-*β*1 induces changes in the permeability of the NPC that can be interpreted as the conformational changes in the FG nup layer. Eibauer et al., Nature Comm, (2015) found that structural transitions at the nucleoplasmic ring of the NPC resemble the transitions of Nup153 between compact conformation when bound to a transport receptor and an extended conformation in the unbound state. Gratton and coworkers showed that Nup153 is likely to undergo long range conformational changes during transport – Cardarelli et al., PNAS, (2012).

We have revised the Abstract and the Introduction to more accurately reflect these points.

*6) Figure 1 is rather crudely rendered, and does not accurately represent the positions of the FG nups indicated. Krull et al., 2004 can be used for Nups 98 and 153, while recent crystallographic work (Chug et al., 2015) indicates the position of Nup62 to be symmetric and immediately adjacent to Nup93 (Krull et al., 2004). Also, the lower two panels as they stand don't really help readers understand what you call the "experimental situation." This figure should be polished further.*

We have improved the presentation and revised the caption of Figure 1. Depiction of the NPC is only schematic with the purpose of illustrating the NPC geometry and architecture. Nevertheless, we agree about the importance of keeping the depiction as accurate as possible, and revised the figure accordingly. We use Chatel et al. (2012) to inform our rendering of the approximate locations of the FG nups on the NPC scaffold. We have added this reference and the references suggested by the reviewer and the editors.

*7) For Figure 2, it can be guessed from the context that different curves correspond to different* χ*s, but it's better to specify which curve correspond to what value (or add some direction like an arrow with "increasing* χ*," etc.*

We added an arrow indicating the direction of increasing *χ_cr_*, and now provide the values of *χ_cr_* in the caption.

*8) For Figure 3, please consider omitting "experimental evidence of cohesiveness" from the figure title. Strictly speaking, it should go something like "polymer model based on cohesion theory fits well with the data" and it's not really evidence for cohesiveness.*

We revised the figure title.